# Resource-rational account of sequential effects in human prediction

Arthur Prat-Carrabin[1,2]*[†], Florent Meyniel[3,4], Rava Azeredo da Silveira[2,5,6]

[1]Department of Economics, Columbia University, New York, United States; [2]Laboratoire de Physique de l'École Normale Supérieure, ENS, Université PSL, CNRS, Sorbonne Université, Université de Paris, Paris, France; [3]Cognitive Neuroimaging Unit, Institut National de la Santé et de la Recherche Médicale, Commissariat à l'Energie Atomique et aux Energies Alternatives, Centre National de la Recherche Scientifique, Université Paris-Saclay, NeuroSpin center, Gif-sur-Yvette, France; [4]Institut de neuromodulation, GHU Paris, Psychiatrie et Neurosciences, Centre Hospitalier Sainte-Anne, Pôle Hospitalo-Universitaire 15, Université Paris Cité, Paris, France; [5]Institute of Molecular and Clinical Ophthalmology Basel, Basel, Switzerland; [6]Faculty of Science, University of Basel, Basel, Switzerland

*For correspondence:
arthurpc@fas.harvard.edu

Present address: [†]Department of Psychology, Harvard University, Cambridge, United States

Competing interest: The authors declare that no competing interests exist.

**Abstract** An abundant literature reports on 'sequential effects' observed when humans make predictions on the basis of stochastic sequences of stimuli. Such sequential effects represent departures from an optimal, Bayesian process. A prominent explanation posits that humans are adapted to changing environments, and erroneously assume non-stationarity of the environment, even if the latter is static. As a result, their predictions fluctuate over time. We propose a different explanation in which sub-optimal and fluctuating predictions result from cognitive constraints (or costs), under which humans however behave rationally. We devise a framework of costly inference, in which we develop two classes of models that differ by the nature of the constraints at play: in one case the precision of beliefs comes at a cost, resulting in an exponential forgetting of past observations, while in the other beliefs with high predictive power are favored. To compare model predictions to human behavior, we carry out a prediction task that uses binary random stimuli, with probabilities ranging from 0.05 to 0.95. Although in this task the environment is static and the Bayesian belief converges, subjects' predictions fluctuate and are biased toward the recent stimulus history. Both classes of models capture this 'attractive effect', but they depart in their characterization of higher-order effects. Only the precision-cost model reproduces a 'repulsive effect', observed in the data, in which predictions are biased away from stimuli presented in more distant trials. Our experimental results reveal systematic modulations in sequential effects, which our theoretical approach accounts for in terms of rationality under cognitive constraints.

## Editor's evaluation

This valuable work addresses a long-standing empirical puzzle from a new computational perspective. The authors provide convincing evidence that attractive and repulsive sequential effects in perceptual decisions may emerge from rational choices under cognitive resource constraints rather than adjustments to changing environments. It is relevant to understanding how people represent uncertain events in the world around them and make decisions, with broad applications to economic behavior.

## Introduction

In many situations of uncertainty, some outcomes are more probable than others. Knowing the probability distributions of the possible outcomes provides an edge that can be leveraged to improve and speed up decision making and perception (*Summerfield and de Lange, 2014*). In the case of choice reaction-time tasks, it was noted in the early 1950s that human reactions were faster when responding to a stimulus whose probability was higher (*Hick, 1952*; *Hyman, 1953*). In addition, faster responses were obtained after a repetition of a stimulus (i.e., when the same stimulus was presented twice in a row), even in the case of serially-independent stimuli (i.e., when the preceding stimulus carried no information on subsequent ones; *Hyman, 1953*; *Bertelson, 1965*). The observation of this seemingly suboptimal behavior has motivated in the following decades a profuse literature on 'sequential effects', i.e., on the dependence of reaction times on the recent history of presented stimuli (*Kornblum, 1967*; *Soetens et al., 1985*; *Cho et al., 2002*; *Yu and Cohen, 2008*; *Wilder et al., 2009*; *Jones et al., 2013*; *Zhang et al., 2014*; *Meyniel et al., 2016*). These studies consistently report a *recency effect* whereby the more often a simple pattern of stimuli (e.g. a repetition) is observed in recent stimulus history, the faster subjects respond to it. In tasks in which subjects are asked to make predictions about sequences of random binary events, sequential effects are also observed and they have given rise since the 1950s to a rich literature (*Jarvik, 1951*; *Edwards, 1961*; *McClelland and Hackenberg, 1978*; *Matthews and Sanders, 1984*; *Gilovich et al., 1985*; *Ayton and Fischer, 2004*; *Burns and Corpus, 2004*; *Croson and Sundali, 2005*; *Bar-Eli et al., 2006*; *Oskarsson et al., 2009*; *Plonsky et al., 2015*; *Plonsky and Erev, 2017*; *Gökaydin and Ejova, 2017*).

Sequential effects are intriguing: why do subjects change their behavior as a function of the recent past observations when those are in fact irrelevant to the current decision? A common theoretical account is that humans infer the statistics of the stimuli presented to them, but because they usually live in environments that change over time, they may believe that the process generating the stimuli is subject to random changes even when it is in fact constant (*Yu and Cohen, 2008*; *Wilder et al., 2009*; *Zhang et al., 2014*; *Meyniel et al., 2016*). Consequently, they may rely excessively on the most recent stimuli to predict the next ones. In several studies, this was heuristically modeled as a 'leaky integration' of the stimuli, that is, an exponential discounting of past observations (*Cho et al., 2002*; *Yu and Cohen, 2008*; *Wilder et al., 2009*; *Jones et al., 2013*; *Meyniel et al., 2016*). Here, instead of positing that subjects hold an incorrect belief on the dynamics of the environment and do not learn that it is stationary, we propose a different account, whereby a cognitive constraint is hindering the inference process and preventing it from converging to the correct, constant belief about the unchanging statistics of the environment. This proposal calls for the investigation of the kinds of choice patterns and sequential effects that would result from different cognitive constraints at play during inference.

We derive a framework of constrained inference, in which a cost hinders the representation of belief distributions (posteriors). This approach is in line with a rich literature that views several perceptual and cognitive processes as resulting from a constrained optimization: the brain is assumed to operate optimally, but within some posited limits on its resources or abilities. The 'efficient coding' hypothesis in neuroscience (*Ganguli and Simoncelli, 2016*; *Wei and Stocker, 2015*; *Wei and Stocker, 2017*; *Prat-Carrabin and Woodford, 2021c*) and the 'rational inattention' models in economics (*Sims, 2003*; *Woodford, 2009*; *Caplin et al., 2019*; *Gabaix, 2017*; *Azeredo da Silveira and Woodford, 2019*; *Azeredo da Silveira et al., 2020*) are examples of this approach, which has been called 'resource-rational analysis' (*Griffiths et al., 2015*; *Lieder and Griffiths, 2019*). Here, we investigate the proposal that human inference is resource-rational, i.e., optimal under a cost. As for the nature of this cost, we consider two natural hypotheses: first, that a higher precision in belief is harder for subjects to achieve, and thus that more precise posteriors come with higher costs; and second, that unpredictable environments are difficult for subjects to represent, and thus that they entail higher costs. Under the first hypothesis, the cost is a function of the belief held, while under the second hypothesis the cost is a function of the inferred environment. We show that the precision cost predicts 'leaky integration': in the resulting inference process, remote observations are discarded. Crucially, beliefs do not converge but fluctuate instead with the recent stimulus history. By contrast, under the unpredictability cost, the inference process does converge, although not to the correct (Bayesian) posterior, but rather to a posterior that implies a biased belief on the temporal structure of the stimuli. In both cases, sequential effects emerge as the result of a constrained inference process.

We examine experimentally the degree to which the models derived from our framework account for human behavior, with a task in which we repeatedly ask subjects to predict the upcoming stimulus in sequences of Bernoulli-distributed stimuli. Most studies on sequential effects only consider the equiprobable case, in which the two stimuli have the same probability. However, the models we consider here are more general than this singular case and they apply to the entire range of stimulus probability. We thus manipulate in separate blocks of trials the stimulus generative probability (i.e., the Bernoulli probability that parameterizes the stimulus) to span the range from 0.05 to 0.95 by increments of 0.05. This enables us to examine in detail the behavior of subjects in a large gamut of environments from the singular case of an equiprobable, maximally-uncertain environment (with a probability of 0.5 for both stimuli) to the strongly-biased, almost-certain environment in which one stimulus occurs with probability 0.95.

To anticipate on our results, the predictions of subjects depend on the stimulus generative probability, but also on the history of stimuli. We examine whether the occurrence of a stimulus, in past trials, increase the proportion of predictions identical to this stimulus ('attractive effect'), or whether it decreases this proportion ('repulsive effect'). The two costs presented above reproduce qualitatively the main patterns in subjects' data, but they make distinct predictions as to the modulations of the recency effect as a function of the history of stimuli, beyond the last stimulus. We show that the responses of subjects exhibit an elaborate, and at times counter-intuitive, pattern of attractive and repulsive effects, and we compare these to the predictions of our models. Our results suggest that the brain infers a stimulus generative probability, but under a constraint on the precision of its internal representations; the inferred generative process may be more general than the actual one, and include higher-order statistics (e.g. transition probabilities), in contrast with the Bernoulli-distributed stimulus used in the experiment.

We present the behavioral task and we examine the predictions of subjects — in particular, how they vary with the stimulus generative probability, and how they depend, at each trial, on the preceding stimulus. We then introduce our framework of inference under constraint, and the two costs we consider, from which we derive two families of models. We examine the behavior of these models and the extent to which they capture the behavioral patterns of subjects. The models make different qualitative predictions about the sequential effects of past observations, which we confront to subjects' data. We find that the predictions of subjects are qualitatively consistent with a model of inference of conditional probabilities, in which precise posteriors are costly.

## Results
### Subjects' predictions of a stimulus increase with the stimulus probability

In a computer-based task, subjects are asked to predict which of two rods the lightning will strike. On each trial, the subject first selects by a key press the left- or right-hand-side rod presented on screen. A lightning symbol (which is here the stimulus) then randomly strikes either of the two rods. The trial is a success if the lightning strikes the rod selected by the subject (*Figure 1a*). The location of the lightning strike (left or right) is a Bernoulli random variable whose parameter $p$ (the stimulus generative probability) we manipulate across blocks of 200 trials: in each block, $p$ is a multiple of 0.05 chosen between 0.05 and 0.95. Changes of block are explicitly signaled to the subjects: each block is presented as a different town exposed to lightning strikes. The subjects are not told that the locations of the strikes are Bernoulli-distributed (in fact no information is given to them regarding how the locations are determined). Moreover, in order to capture the 'stationary' behavior of subjects, which presumably prevails after ample exposure to the stimulus, each block is preceded by 200 passive trials in which the stimuli (sampled with the probability chosen for the block) are successively shown with no action from the subject (*Figure 1b*); this is presented as a 'useful track record' of lightning strikes in the current town. (To verify the stationarity of subjects' behavior, we compare their responses in the first and second halves of the 200 trials in which they are asked to make predictions. In most cases we find no significant differences. See Appendix.) We provide further details on the task in Methods.

The behavior of subjects varies with the stimulus generative probability, $p$. In our analyses, we are interested in how the subjects' predictions of an event (left or right strike) vary with the probability of this event, regardless of its nature (left or right). Thus, for instance, we would like to pool together

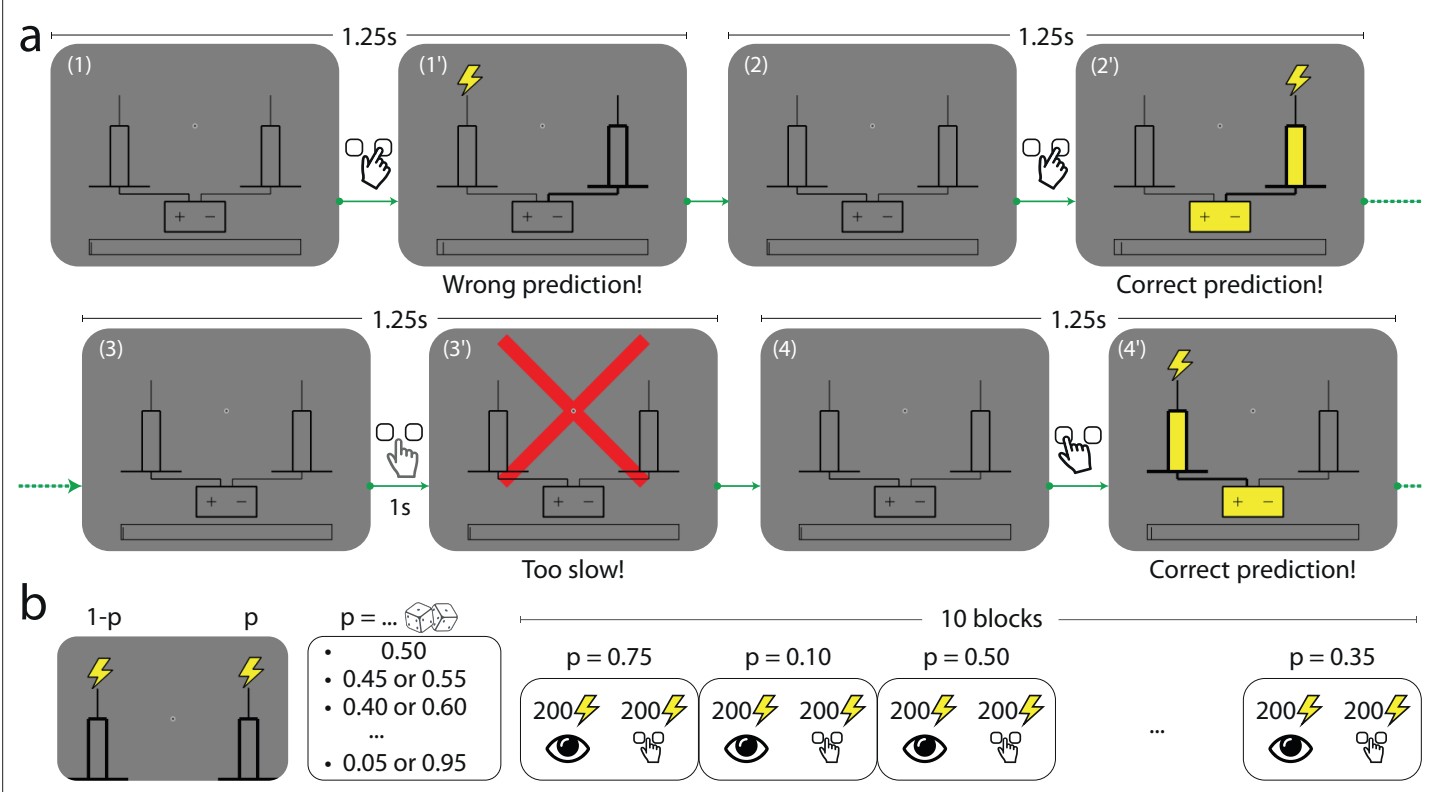

**Figure 1.** The Bernoulli sequential prediction task. (**a**) In each successive trial, the subject is asked to predict which of two rods the lightning will strike. (1) A trial begins. (1') The subject chooses the right-hand-side rod (bold lines), but the lightning strikes the left one (this feedback is given immediately after the subject makes a choice). (2) 1.25 s after the beginning of the preceding trial, a new trial begins. (2') The subject chooses the right-hand-side rod, and this time the lightning strikes the rod chosen by the subject (immediate feedback). The rod and the connected battery light up (yellow), indicating success. (3) A new trial begins. (3') If after 1 s the subject has not made a prediction, a red cross bars the screen and the trial ends. (4) A new trial begins. (4') The subject chooses the left-hand-side rod, and the lightning strikes the same rod. In all cases, the duration of a trial is 1.25 s. (**b**) In each block of trials, the location of the lightning strike is a Bernoulli random variable with parameter $p$, the stimulus generative probability. Each subject experiences 10 blocks of trials. The stimulus generative probability for each block is chosen randomly among the 19 multiples of 0.05 ranging from 0.05 to 0.95, with the constraint that if $p$ is chosen for a given block, neither $p$ nor $1 - p$ can be chosen in the subsequent blocks; as a result for any value $p$ among these 19 probabilities spanning the range from 0.05 to 0.95, there is one block in which one of the two rods receives the lightning strike with probability $p$. Within each block the first 200 trials consist in passive observation and the 200 following trials are active trials (depicted in panel a).

the trials in which a subject makes a rightward prediction when the probability of a rightward strike is 0.7, and the trials in which a subject makes a leftward prediction when the probability of a leftward strike is also 0.7. Therefore, throughout the paper, we do not discuss whether subjects predict 'right' or 'left', and instead we discuss whether they predict the event 'A' or the complementary event 'B': in different blocks of trials, A (and similarly B) may refer to different locations; but importantly, B always corresponds to the location opposite to A, and $p$ denotes the probability of A (thus B has probability $1 - p$). This allows us, given a probability $p$, to pool together the responses obtained in blocks of trials in which one of the two locations has probability $p$. One advantage of this pooling is that it reduces the noise in data. Looking at the unpooled data, however, does not change our conclusions; see Appendix.

Turning to the behavior of subjects, we denote by $\bar{p}(A)$ the proportion of trials in which a subject predicts the event A. In the equiprobable condition ($p = 0.5$), the subjects predict either side on about half the trials ($\bar{p}(A) = .496$, subjects pooled; standard error of the mean (sem): 0.008; p-value of t-test of equality with 0.5: 0.59). In the non-equiprobable conditions, the optimal behavior is to predict A on none of the trials ($\bar{p}(A) = 0$) if $p < 0.5$, or on all trials ($\bar{p}(A) = 1$) if $p > 0.5$. The proportion of predictions A adopted by the subjects also increases as a function of the stimulus generative probability (Pearson correlation coefficient between $p$ and $\bar{p}(A)$, subjects pooled: .97; p-value: 3.3e-6; correlation between

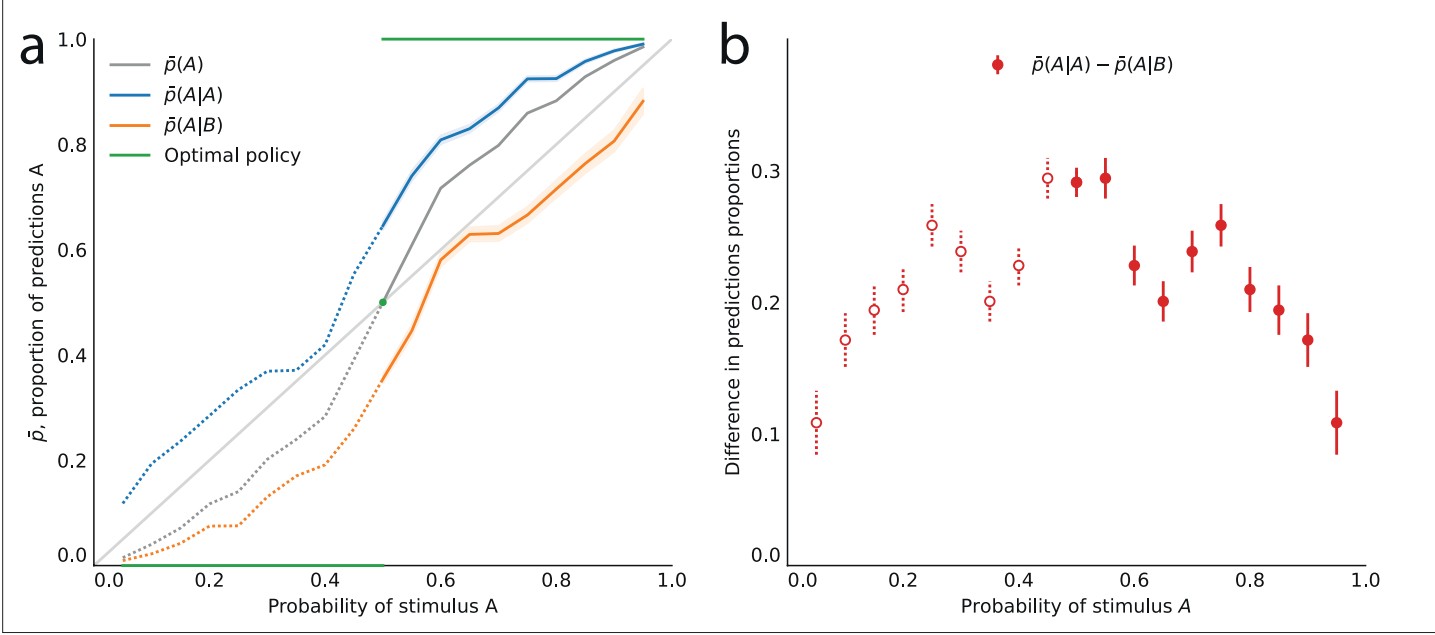

**Figure 2.** Across all stimulus generative probabilities, subjects are more likely than average to make a prediction equal to the preceding observation. (**a**) Proportion of predictions A in subjects' pooled responses as a function of the stimulus generative probability, conditional on observing an A (blue line) or a B (orange line) on the preceding trial, and unconditional (grey line). The widths of the shaded areas indicate the standard error of the mean (n = 178 to 3603). (**b**) Difference between the proportion of predictions A conditional on the preceding observation being an A, and the proportion of predictions A conditional on the preceding observation being a B. This difference is positive across all stimulus generative probabilities, that is, observing an A at the preceding trial increases the probability of predicting an A (p-values of Fisher's exact tests, with Bonferroni-Holm-Šidák correction, are all below 1e-13). Bars are twice the square root of the sum of the two squared standard errors of the means (for each point, total n: 3582 to 3781). The binary nature of the task results in symmetries in this representation of data: in panel (**a**) the grey line is symmetric about the middle point and the blue and orange lines are reflections of each other, and in panel (**b**) the data is symmetric about the middle probability, 0.5. For this reason, for values of the stimulus generative probability below 0.5 we show the curves in panel (**a**) as dotted lines, and the data points in panel (**b**) as white dots with dotted error bars.

the 'logits', $\ln \frac{p}{1-p}$: 0.994, p-value: 5.7e-9.), but not as steeply: it lies between the stimulus generative probability $p$, and the optimal response 0 (if $p < 0.5$) or 1 (if $p > 0.5$; *Figure 2a*).

## First-order sequential effects: attractive influence of the most recent stimulus on subjects' predictions

The sequences presented to subjects correspond to independent, Bernoulli-distributed random events. Having shown that the subjects' predictions follow (in a non-optimal fashion) the stimulus generative probability, we now test whether they also exhibit the non-independence of consecutive trials featured by the Bernoulli process. Under this hypothesis and in the stationary regime, the proportion of predictions A conditional on the preceding stimulus being A, $\bar{p}(A|A)$, should be no different than the proportion of predictions A conditional on the preceding stimulus being B, $\bar{p}(A|B)$. (Here and below, $\bar{p}(X|Y)$ denotes the proportion of predictions X conditional on the preceding observation being Y, and not on the preceding response being Y. For the possibility that subjects' responses depend on the preceding response, see Methods.)

In other words, conditioning on the preceding stimulus should have no effect. In subjects' responses, however, these two conditional proportions are markedly different for all stimulus generative probabilities (Fisher exact test, subjects pooled: all p-values < 1e-10; *Figure 2a*). Both quantities increase as a function of the stimulus generative probability, but the proportions of predictions A conditional on an A are consistently greater than the proportions of predictions A conditional on a B, i.e., $\bar{p}(A|A) - \bar{p}(A|B) > 0$ (*Figure 2b*). (We note that because the stimulus is either A or B, it follows that, symmetrically, the proportions of predictions B conditional on a B are consistently greater than the proportions of predictions B conditional on an A.) In other words, the preceding stimulus has an

'attractive' sequential effect. In addition, this attractive sequential effect seems stronger for values of the stimulus generative probability closer to the equiprobable case (p = 0.5), and to decrease for more extreme values ($p$ closer to 0 or to 1; *Figure 2b*). The results in *Figure 2* are obtained by pooling together the responses of the subjects. Results derived from an across-subjects analysis are very similar; see Appendix.

## A framework of costly inference

The attractive effect of the preceding stimulus on subjects' responses suggests that the subjects have not correctly inferred the Bernoulli statistics of the process generating the stimuli. We investigate the hypothesis that their ability to infer the underlying statistics of the stimuli is hampered by cognitive constraints. We assume that these constraints can be understood as a cost, bearing on the representation, by the brain, of the subject's beliefs about the statistics. Specifically, we derive an array of models from a framework of inference under costly posteriors (*Prat-Carrabin et al., 2021a*), which we now present. We consider a model subject who is presented on each trial $t$ with a stimulus $x_t \in \{0, 1\}$ (where 0 and 1 encode for B and A, respectively) and who uses the sequence of stimuli $x_{1:t} = (x_1, \ldots, x_t)$ to infer the stimulus statistics, over which she holds the belief distribution $\hat{P}_t$. A Bayesian observer equipped with this belief $\hat{P}_t$ and observing a new observation $x_{t+1}$ would obtain its updated belief $P_{t+1}$ through Bayes' rule. However, a cognitive cost $C(P)$ hinders our model subject's ability to represent probability distributions $P$. Thus, she approximates the posterior $P_{t+1}$ through another distribution $\hat{P}_{t+1}$ that minimizes a loss function $L$ defined as

$$L(\hat{P}_{t+1}) = D(\hat{P}_{t+1}; P_{t+1}) + \lambda C(\hat{P}_{t+1}), \tag{1}$$

where $D$ is a measure of distance between two probability distributions, and $\lambda \geq 0$ is a coefficient specifying the relative weight of the cost. (We are not proposing that subjects actively minimize this quantity, but rather that the brain's inference process is an effective solution to this optimization problem.) Below, we use the Kullback-Leibler divergence for the distance (i.e. $D(\hat{P}_{t+1}; P_{t+1}) = D_{KL}(\hat{P}_{t+1} \| P_{t+1})$). If $\lambda = 0$, the solution to this minimization problem is the Bayesian posterior; if $\lambda \neq 0$, the cost distorts the Bayesian solution in ways that depend on the form of the cost borne by the subject (we detail further below the two kinds of costs we investigate).

In our framework, the subject assumes that the $m$ preceding stimuli ($x_{t-m+1:t}$ with $m \geq 0$) and a vector of parameters $q$ jointly determine the distribution of the stimulus at trial $t + 1$, $p(x_{t+1}|x_{t-m+1:t}, q)$. Although in our task the stimuli are Bernoulli-distributed (thus they do not depend on preceding stimuli) and a single parameter determines the probability of the outcomes (the stimulus generative probability), the subject may admit the possibility that more complex mechanisms govern the statistics of the stimuli, for example transition probabilities between consecutive stimuli. Therefore, the vector $q$ may contain more than one parameter and the number $m$ of preceding stimuli assumed to influence the probability of the following stimulus, which we call the 'Markov order', may be greater than 0.

Below, we call 'Bernoulli observer' any model subject who assumes that the stimuli are Bernoulli-distributed ($m = 0$); in this case the vector $q$ consists of a single parameter that determines the probability of observing A, which we also denote by $q$ for the sake of concision. The bias and variability in the inference of the Bernoulli observer is studied in *Prat-Carrabin et al., 2021a*. We call 'Markov observer' any model subject who posits that the probability of the stimulus depends on the preceding stimuli ($m > 0$). In this case, the vector $q$ contains the $2^m$ conditional probabilities of observing A after observing each possible sequence of $m$ stimuli. For instance, with $m = 1$ the vector $q$ is the pair of parameters $(q_A, q_B)$ denoting the probabilities of observing a stimulus A after observing, respectively, a stimulus A and a stimulus B. In the absence of a cost, the belief over the parameter(s) eventually converges towards the parameter vector that is consistent with the generative Bernoulli statistics governing the stimulus (except if the prior precludes this parameter vector). Below, we assume a uniform prior.

To understand how the costs contort the inference process, it is useful to have in mind the solution to the 'unconstrained' inference problem (with $\lambda = 0$), i.e., the Bayesian posterior, which we denote by $P_t^*(q)$. In the case of a Bernoulli observer ($m = 0$), after $t$ trials, the Bayesian posterior is a Beta distribution,

$$P_t^*(q) \propto q^{n_t^A}(1 - q)^{n_t^B}, \tag{2}$$

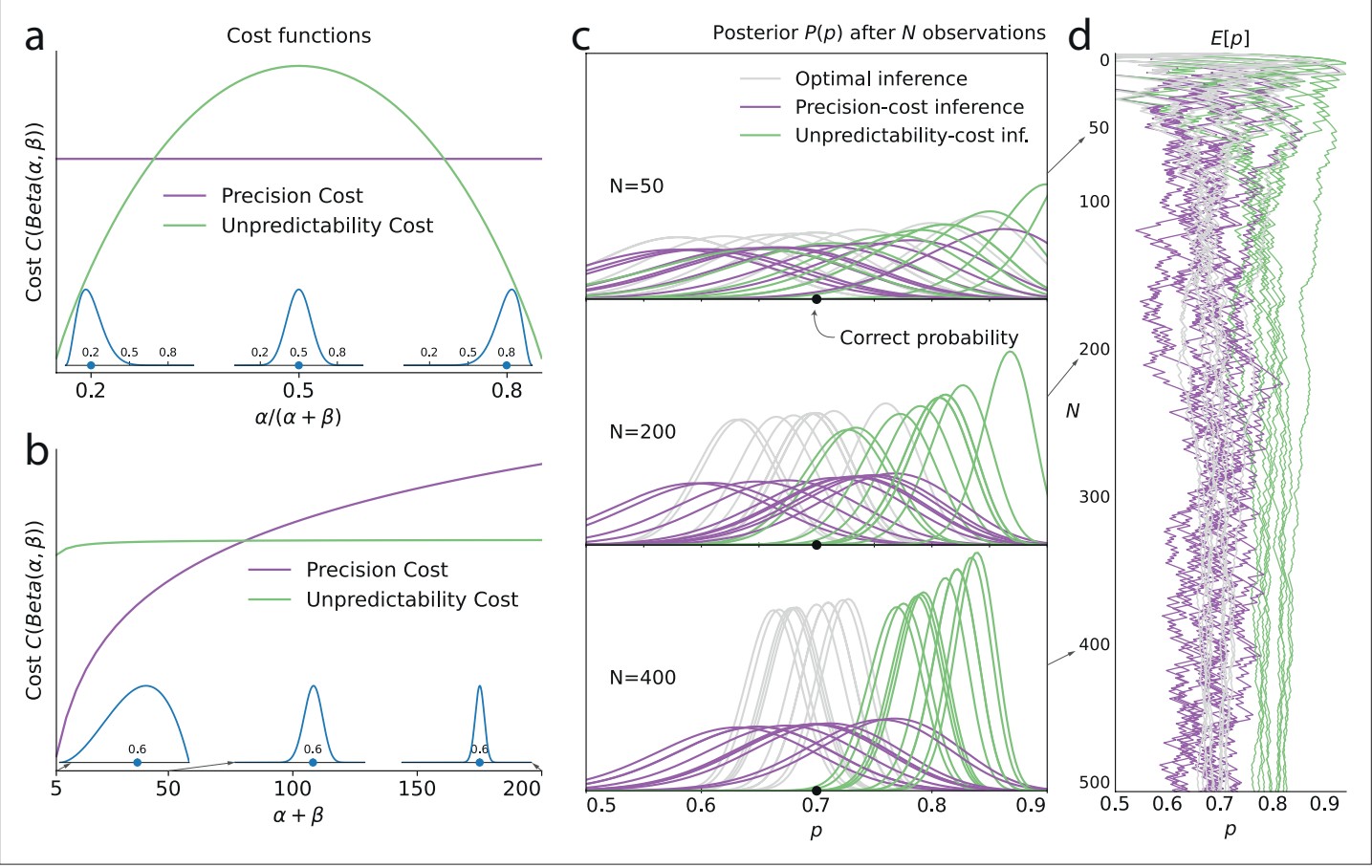

**Figure 3.** Illustration of the Bernoulli-observer models, with unpredictability and precision costs. (**a**) Precision cost (purple) and unpredictability cost (green lines) of a Beta distribution with parameters $\alpha$ and $\beta$, as functions of the mean of the distribution, $\alpha/(\alpha + \beta)$, and keeping the entropy constant. The precision cost is the negative of the entropy and it is thus constant, regardless of the mean of the distribution. The unpredictability cost is larger when the mean of the distribution is closer to 0.5 (i.e. when unpredictable environments are likely, under the distribution). *Insets:* Beta distributions with mean 0.2, 0.5, and 0.8, and constant entropy. (**b**) Costs as functions of the sample size parameter, $\alpha + \beta$. A larger sample size implies a higher precision and lower entropy, thus the precision cost increases as a function of the sample size, whereas the unpredictability cost is less sensitive to changes in this parameter. *Insets:* Beta distributions with mean 0.6 and sample size parameter, $\alpha + \beta$, equal to 5, 50, and 200. (**c**) Posteriors $P(p)$ of an optimal observer (gray), a precision-cost observer (purple) and an unpredictability-cost observer (green lines), after the presentation of ten sequences of N = 50, 200, and 400 observations sampled from a Bernoulli distribution of parameter 0.7. The posteriors of the optimal observer narrow as evidence is accumulated, and the different posteriors obtained after different sequences of observations are drawn closer to each other and to the correct probability. The posteriors of the unpredictability-cost observer also narrow and group together, but around a probability larger (less unpredictable) than the correct probability. Precise distributions are costly to the precision-cost observer and thus the posteriors do not narrow after long sequences of observations. Instead, the posteriors fluctuate with the recent history of the stimuli. (**d**) Expected probability $E[p]$ resulting from the inference. The optimal observer (gray) converges towards the correct probability; the unpredictability-cost observer (green) converges towards a biased (larger) probability; and the precision-cost observer (purple lines) does not converge, but instead fluctuates with the history of the stimuli.

where $n_t^X$ is the number of stimuli $X$ observed up to trial $t$, that is, $n_t^A = \sum_{i=1}^{t} x_i$, and $n_t^B = \sum_{i=1}^{t} (1 - x_i)$. As more evidence is accumulated, the Bayesian posterior gradually narrows and converges towards the value of the stimulus generative probability (*Figure 3c and d*, grey lines).

The ways in which the Bayesian posterior is distorted, in our models, depend on the nature of the cost that weighs on the inference process. Although many assumptions could be made on the kind of constraint that hinders human inference, and on the cost it would entail in our framework, here we examine two costs that stem from two possible principles: that the cost is a function of the beliefs held by the subject, or that it is a function of the environment that the subject is inferring. We detail, below, these two costs.

## Precision cost

A first hypothesis about the inference process of subjects is that the brain mobilizes resources to represent probability distributions, and that more 'precise' distributions require more resources. We write the cost associated with a distribution, $\hat{P}(q)$, as the negative of its entropy,

$$C_p(\hat{P}) = -H[\hat{P}(q)] = \int \hat{P}(q) \ln \hat{P}(q) \, \mathrm{d}q, \tag{3}$$

which is a measure of the amount of certainty in the distribution. Wider (less concentrated) distributions provide less information about the probability parameter and are thus less costly than narrower (more concentrated) distributions (*Figure 3b*). As an extreme case, the uniform distribution is the least costly.

With this cost, the loss function (*Equation 1*) is minimized by the distribution equal to the product of the prior and the likelihood, raised to the exponent $1/(\lambda + 1)$, and normalized, i.e.,

$$\hat{P}_{t+1}(q) \propto \left[ \hat{P}_t(q) p(x_{t+1}|x_{t-m+1:t}, q) \right]^{1/(\lambda+1)}. \tag{4}$$

Since $\lambda$ is strictly positive, the exponent is positive and lower than 1. As a result, the solution 'flattens' the Bayesian posterior, and in the extreme case of an unbounded cost ($\lambda \to \infty$) the posterior is the uniform distribution.

Furthermore, in the expression of our model subject's posterior, the likelihood $p(x_{t+1}|x_{t-m+1:t}, q)$ is raised after $k$ trials to the exponent $1/(\lambda + 1)^{k+1}$, it thus decays to zero as the number $k$ of new stimuli increases. One can interpret this effect as gradually forgetting past observations. Specifically, we recover the predictions of leaky-integration models, in which remote patterns in the sequence of stimuli are discounted through an exponential filter (*Yu and Cohen, 2008*; *Meyniel et al., 2016*); here, we do not posit the gradual forgetting of remote observations, but instead we derive it as an optimal solution to a problem of constrained inference. We illustrate leaky integration in the case of a Bernoulli observer ($m = 0$): in this case, the posterior after $t$ trials, $\hat{P}_t(q)$, is a Beta distribution,

$$\hat{P}_t(q) \propto q^{\tilde{n}_t^A} (1 - q)^{\tilde{n}_t^B}, \tag{5}$$

where $\tilde{n}_t^A$ and $\tilde{n}_t^B$ are exponentially-filtered counts of the number of stimuli A and B observed up to trial $t$, i.e.,

$$\tilde{n}_t^A = \sum_{k=0}^{t-1} \left( \frac{1}{\lambda + 1} \right)^k x_{t-k}, \text{and } \tilde{n}_t^B = \sum_{k=0}^{t-1} \left( \frac{1}{\lambda + 1} \right)^k (1 - x_{t-k}). \tag{6}$$

In other words, the solution to the constrained inference problem, with the precision cost, is similar to the Bayesian posterior (*Equation 2*), but with counts of the two stimuli that gradually 'forget' remote observations (in the absence of a cost, that is, $\lambda = 0$, we have $\tilde{n}_t^A = n_t^A$ and $\tilde{n}_t^B = n_t^B$, and thus we recover the Bayesian posterior). As a result, these counts fluctuate with the recent history of the stimuli. Consequently, the posterior $\hat{P}_t(q)$ is dominated by the recent stimuli: it does not converge, but instead fluctuates with the recent stimulus history (*Figure 3c and d*, purple lines; compare with the green and gray lines). Hence, this model implies predictions about subsequent stimuli that depend on the stimulus history, i.e., it predicts sequential effects.

## Unpredictability cost

A different hypothesis is that the subjects favor, in their inference, parameter vectors $q$ that correspond to more predictable outcomes. We quantify the outcome unpredictability by the Shannon entropy (*Shannon, 1948*) of the outcome implied by the vector of parameters $q$, which we denote by $H(X; q)$. (In the Bernoulli-observer case, $H(X; q) = -q \ln q - (1 - q) \ln(1 - q)$; for the Markov-observer cases, see Methods.) The cost associated with the distribution $\hat{P}(q)$ is the expectation of this quantity averaged over beliefs, i.e.,

$$C_u(\hat{P}) = \mathbb{E}_{\hat{P}}[H(X; q)] = \int H(X; q) \hat{P}(q) \mathrm{d}q, \tag{7}$$

which we call the 'unpredictability cost'. For a Bernoulli observer, a posterior concentrated on extreme values of the Bernoulli parameter (toward 0 or 1), thus representing more predictable environments, comes with a lower cost than a posterior concentrated on values of the Bernoulli parameter close to 0.5, which correspond to the most unpredictable environments (**Figure 3a**).

After $t$ trials, the loss function (**Equation 1**) under this cost is minimized by the posterior

$$\hat{P}_t(q) \propto P_t^*(q)e^{-t\lambda H(X;q)}, \tag{8}$$

i.e., the product of the Bayesian posterior, which narrows with $t$ around the stimulus generative probability, and of a function that is larger for values of $q$ that imply less entropic (i.e. more predictable) environments (see Methods). In short, with the unpredictability cost the model subject's posterior is 'pushed' towards less entropic values of $q$.

In the Bernoulli case ($m = 0$), the posterior after $t$ stimuli has a global maximum, $q^*(n_t/t)$, that depends on the proportion $n_t/t$ of stimuli A observed up to trial $t$. As the number of presented stimuli $t$ grows, the posterior $\hat{P}_t$ becomes concentrated around this maximum. The proportion $n_t/t$ naturally converges to the stimulus generative probability, $p$, thus our subject's inference converges towards the value $q^*(p)$ which is different from the true value $p$, in the non-equiprobable case ($p \neq .5$). The equiprobable case ($p = .5$) is singular, in that with a weak cost ($\lambda < 1$) the inferred probability is unbiased ($q^*(p) = .5$), while with a strong cost ($\lambda > 1$) the inferred probability does not converge but instead alternates between two values above and below 0.5; see **Prat-Carrabin et al., 2021a**. In other words, except in the equiprobable case, the inference converges but it is biased, i.e., the posterior peaks at an incorrect value of the stimulus generative probability (**Figure 3c and d**, green lines). This value is closer to the extremes (0 and 1) than the stimulus generative probability, that is, it implies an environment more predictable than the actual one (**Figure 3d**).

In the case of a Markov observer ($m > 0$), the posterior also converges to a vector of parameters $q$ which implies not only a bias but also that the conditional probabilities of a stimulus A (conditioned on different stimulus histories) are not equal. The prediction of the next stimulus being A on a given trial depends on whether the preceding stimulus was A or B: this model therefore predicts sequential effects. We further examine below the behavior of this model in the cases of a Bernoulli observer and of different Markov observers. We refer the reader interested in more details on the Markov models, including their mathematical derivations, to the Methods section.

In short, with the unpredictability-cost models, when $p \neq 0.5$, the inference process converges to an asymptotic posterior $q^*(p)$ which does not itself depend on the history of the stimulus, but that is biased (**Figure 3c, d**, green lines). In particular, for Markov observers ($m > 0$), the asymptotic posterior corresponds to an erroneous belief about the dependency of the stimulus on the recent stimulus history, which results in sequential effects in behavior.

## Overview of the inference models

Although the two families of models derived from the two costs both potentially generate sequential effects, they do so by giving rise to qualitatively different inference processes. Under the unpredictability cost, the inference converges to a posterior that, in the Bernoulli case ($m = 0$), implies a biased estimate of the stimulus generative probability (**Figure 3d**, green lines), while in the Markov case ($m > 0$) it implies the belief that there are serial dependencies in the stimuli: predictions therefore depend on the recent stimulus history. By contrast, the precision cost prevents beliefs from converging (**Figure 3c**, purple lines). As a result, the subject's predictions vary with the recent stimulus history (**Figure 3d**). This inference process amounts to an exponential discount of remote observations, or equivalently, to the overweighting of recent observations (**Equation 6**).

To investigate in more detail the sequential effects that these two costs produce, we implement two families of inference models derived from the two costs. Each model is characterized by the type of cost (unpredictability cost or precision cost), and by the assumed Markov order ($m$): we examine the case of a Bernoulli observer ($m = 0$) and three cases of Markov observers (with $m = 1, 2$, and $3$). We thus obtain $2 \times 4 = 8$ models of inference. Each of these models has one parameter $\lambda$ controlling the weight of the cost. (We also examine a 'hybrid' model that combines the two costs; see below.)

## Response-selection strategy

We assume that the subject's response on a given trial depends on the inferred posterior according to a generalization of 'probability matching' implemented in other studies (*Battaglia et al., 2011*; *Yu and Huang, 2014*; *Prat-Carrabin et al., 2021b*). In this response-selection strategy, the subject predicts A with the probability $\bar{p}_t^\kappa/(\bar{p}_t^\kappa + (1-\bar{p}_t)^\kappa)$, where $\bar{p}_t$ is the expected probability of a stimulus A derived from the posterior, i.e., $\bar{p}_t \equiv \int p(x_{t+1} = 1|x_{t-m+1:t}, q)\hat{P}_t(q)\mathrm{d}q$. The single parameter $\kappa$ controls the randomness of the response: with $\kappa = 0$ the subject predicts A and B with equal probability; with $\kappa = 1$ the response-selection strategy corresponds to probability matching, that is, the subject predicts A with probability $\bar{p}_t$; and as $\kappa$ increases toward infinity the choices become optimal, that is, the subjects predicts A if the expected probability of observing a stimulus A, $\bar{p}_t$, is greater than 0.5, and predicts B if it is lower than 0.5 (if $\bar{p}_t = 0.5$ the subject chooses A or B with equal probability). In our investigations, we also implement several other response-selection strategies, including one in which subjects have a propensity to repeat their preceding response, or conversely, to alternate; these analyses do not change our conclusions (see Methods).

## Model fitting favors Markov-observer models

Each of our eight models has two parameters: the factor weighting the cost, $\lambda$, and the exponent of the generalized probability-matching, $\kappa$. We fit the parameters of each model to the responses of each subject, by maximizing their likelihoods. We find that 60% of subjects are best fitted by one of the unpredictability-cost models, while 40% are best fitted by one of the precision-cost models. When pooling the two types of cost, 65% of subjects are best fitted by a Markov-observer model. We implement a 'Bayesian model selection' procedure (*Stephan et al., 2009*), which takes into account, for each subject, the likelihoods of all the models (and not only the maximum among them) in order to obtain a Bayesian posterior over the distribution of models in the general population (see Methods). The derived expected probability of unpredictability-cost models is 57% (and 43% for precision-cost models) with an exceedance probability (i.e. probability that unpredictability-cost models are more frequent in the general population) of 78%. The expected probability of Markov-observer models, regardless of the cost used in the model, is 70% (and 30% for Bernoulli-observer models) with an exceedance probability (i.e. probability that Markov-observer models are more frequent in the general population) of 98%. These results indicate that the responses of subjects are generally consistent with a Markov-observer model, although the stimuli used in the experiment are Bernoulli-distributed. As for the unpredictability-cost and the precision-cost families of models, Bayesian model selection does not provide decisive evidence in favor of either model, indicating that they both capture some aspects of the responses of the subjects. Below, we examine more closely the behaviors of the models, and point to qualitative differences between the predictions resulting from each model family.

Before turning to these results, we validate the robustness of our model-fitting procedure with several additional analyses. First, we estimate a confusion matrix to examine the possibility that the model-fitting procedure could misidentify the models which generated test sets of responses. We find that the best-fitting model corresponds to the true model in at least 70% of simulations (the chance level is 12.5%=1/8 models), and actually more than 90% for the majority of models (see Appendix).

Second, we seek to verify whether the best-fitting cost factor, $\lambda$, that we obtain for each subject is consistent across the range of probabilities tested. Specifically, we fit separately the models to the responses obtained in the blocks of trials whose stimulus generative probability was 'medium' (between 0.3 and 0.7, included) on the one hand, and to the responses obtained when the probability was 'extreme' (below 0.3, and above 0.7) on the other hand; and we compare the values of the best-fitting cost factors $\lambda$ in these two cases. More precisely, for the precision-cost family, we look at the inverse of the decay time, $\ln(1 + \lambda)$, which is the inverse of the characteristic time over which the model subject 'forgets' past observations. With both families of models, we find that on a logarithmic scale the parameters in the medium- and extreme-probabilities cases are significantly correlated across subjects (Pearson's $r$, precision-cost models: 0.75, p-value: 1e-4; unpredictability-cost models: $r = 0.47$, p-value: .036). In other words, if a subject is best fitted by a large cost factor in medium-probabilities trials, he or she is likely to be also best fitted by a large cost factor in extreme-probabilities trials. This indicates that our models capture idiosyncratic features of subjects that generalize across conditions instead of varying with the stimulus probability (see Appendix).

Third, as mentioned above we examine a variant of the response-selection strategy in which the subject sometimes repeats the preceding response, or conversely alternates and chooses the other response, instead of responding based on the inferred probability of the next stimulus. This propensity to repeat or alternate does not change the best-fitting inference model of most subjects, and the best-fitting values of the parameters $\lambda$ and $\kappa$ are very stable when allowing or not for this propensity. This analysis supports the results we present here, and speaks to the robustness of the model-fitting procedure (see Methods).

Finally, as the unpredictability-cost family and the precision-cost family of models both seem to capture the responses of a sizable share of the subjects, one might assume that the behavior of most subjects actually fall 'somewhere in between', and would be best accounted for by a hybrid model combining the two costs. In our investigations, we have implemented such a model, whereby the subject's approximate posterior $\hat{P}_t$ results from the minimization of a loss function that includes both a precision cost, with weight $\lambda_p$, and an unpredictability cost, with weight $\lambda_u$ (and the response-selection strategy is the generalized probability matching, with parameter $\kappa$). We do not find that most subjects' responses are better fitted (as measured by the Bayesian Information Criterion *Schwarz, 1978*) by a combination of the two costs: instead, for more than two thirds of subjects, the best-fitting model features just one cost (see Methods). In other words, the two cost seems to capture different aspects of the behavior that are predominant in different subpopulations. Below, we examine the behavioral patterns resulting from each cost type, in comparison with the behavior of the subjects.

## Models of costly inference reproduce the attractive effect of the most recent stimulus

We now examine the behavioral patterns resulting from the models. All the models we consider predict that the proportion of predictions A, $\bar{p}(A)$, is a smooth, increasing function of the stimulus generative probability (when $\lambda < \infty$ and $0 < \kappa < \infty$; *Figure 4a–d*, grey lines), thus we focus, here, on the ability of the models to reproduce the subjects' sequential effects. With the unpredictability-cost model of a Bernoulli observer ($m = 0$), the belief of the model subject, as mentioned above, asymptotically converges in non-equiprobable cases to an erroneous value of the stimulus generative probability (*Figure 3d*, green lines). After a large number of observations (such as the 200 'passive' trials, in our task), the sensitivity of the belief to new observations becomes almost imperceptible; as a result, this model predicts practically no sequential effects (*Figure 4b*), that is, $\bar{p}(A|A) \simeq \bar{p}(A|B)$. With the unpredictability-cost model of a Markov observer (e.g. $m = 1$), the belief of the model subject also converges, but to a vector of parameters $q$ that implies a sequential dependency in the stimulus, that is, $q_A \neq q_B$, resulting in sequential effects in predictions, that is, $\bar{p}(A|A) \neq \bar{p}(A|B)$. The parameter vector $q$ yields a more predictable (less entropic) environment if the probability conditional on the more frequent outcome (say, A) is less entropic than the probability conditional on the less frequent outcome (B). This is the case if the former is greater than the latter, resulting in the inequality $\bar{p}(A|A) > \bar{p}(A|B)$, that is, in sequential effects of the attractive kind (*Figure 4d*). (The case in which B is the more frequent outcome results in the inequality $\bar{p}(B|B) > \bar{p}(B|A)$, i.e., $1 - \bar{p}(A|B) > 1 - \bar{p}(A|A)$, i.e., the same, attractive sequential effects.)

Turning to the precision-cost models, we have noted that in these models the posterior fluctuates with the recent history of the stimuli (*Figure 3c*): as a result, sequential effects are obtained, even with a Bernoulli observer ($m = 0$; *Figure 4a*). The most recent stimulus has the largest weight in the exponentially filtered counts that determine the posterior (*Equation 6*), thus the model subject's prediction is biased towards the last stimulus, that is, the sequential effect is attractive ($\bar{p}(A|A) > \bar{p}(A|B)$). With the traditional probability-matching response-selection strategy (i.e. $\kappa = 1$), the strength of the attractive effect is the same across all stimulus generative probabilities (i.e. the difference $\bar{p}(A|A) - \bar{p}(A|B)$ is constant; *Figure 4a*, dotted lines and light-red dots). With the generalized probability-matching response-selection strategy, if $\kappa > 1$, proportions below and above 0.5 are brought closer to the extremes (0 and 1, respectively), resulting in larger sequential effects for values of the stimulus generative probability closer to 0.5 (*Figure 4a*, solid lines and red dots; the model is simulated with $\kappa = 2.8$, a value representative of the subjects' best-fitting values for this parameter). We also find stronger sequential effects closer to the equiprobable case in subjects' data (*Figure 2b*).

The precision-cost model of a Markov observer ($m = 1$) also predicts attractive sequential effects (*Figure 4c*). While the behavior of the Bernoulli observer (with a precision cost) is determined by two

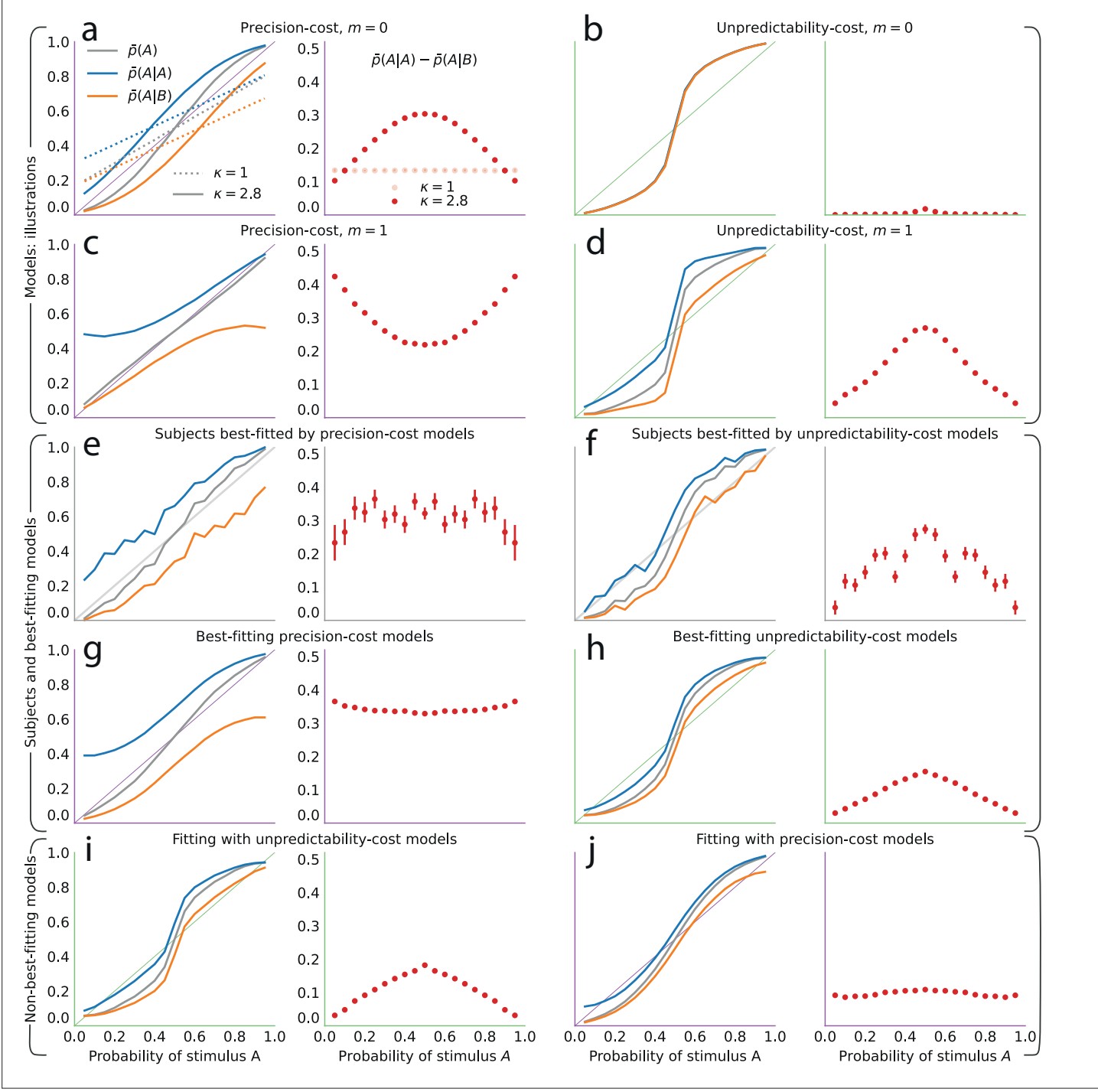

**Figure 4.** The precision-cost and unpredictability-cost models reproduce the subjects' attractive sequential effects. (**a–h**) *Left subpanels:* proportion of predictions A as a function of the stimulus generative probability, conditional on observing A (blue line) or B (orange line) on the preceding trial, and unconditional (grey line). *Right subpanels:* difference between the proportion of predictions A conditional on observing A, and conditional on observing B, $\bar{p}(A|A) - \bar{p}(A|B)$. In all panels this difference is positive, indicating an attractive sequential effect (i.e. observing A at the preceding trial increases the probability of predicting A). (**a–d**) Models with the precision cost (**a,c**) or the unpredictability cost (**b,d**), and with a Bernoulli observer ($m = 0$; **a,b**) or a Markov observer with $m = 1$ (**c,d**). (**a**) Behavior with the generalized probability-matching response-selection strategy with $\kappa = 2.8$ (solid lines, red dots) and with $\kappa = 1$ (dotted lines, light-red dots). (**e,f**) Pooled responses of the subjects best-fitted by a precision-cost model (**e**) or by an unpredictability-cost model (**f**). *Right subpanels:* bars are twice the square root of the sum of the two squared standard errors of the means (for each point, total n: **e**: 1393, f: 2189 to 2388). (**g,h**) Pooled responses of the models that best fit the corresponding subjects in panels (**e,f**). (**i,j**) Pooled responses of the unpredictability-cost models (**i**) and of the precision-cost models (**j**), fitted to the subjects best-fitted, respectively, by precision-cost models, and by unpredictability-cost models.

exponentially-filtered counts of the two possible stimuli (*Equation 6*), that of the Markov observer with $m = 1$ depends on four exponentially filtered counts of the four possible pairs of stimuli. After observing a stimulus B, the belief that the following stimulus should be A or B is determined by the exponentially filtered counts of the pairs BA and BB. If $p$ is large, i.e., if the stimulus B is infrequent, then the BA and BB pairs are also infrequent and the corresponding counts are close to zero: the model subject thus behaves as if only very little evidence had been observed about the transitions B to A and B to B in this case, resulting in a proportion of predictions A conditional on a preceding B, $\bar{p}(A|B)$, close to 0.5 (*Figure 4c*, orange line). Consequently, the sequential effects are stronger for values of the stimulus generative probabilities closer to the extreme (*Figure 4c*, red dots).

Both families of costs are thus able to produce attractive sequential effects, albeit with some qualitative differences. (In *Figure 4a–d* we show the behaviors resulting from the two costs for a Bernoulli observer and a Markov observer of order $m = 1$; the Markov observers of higher order exhibit qualitatively similar behaviors; see Methods.) As the model fitting indicates that different groups of subjects are best fitted by models belonging to the two families, we examine separately the behaviors of the subjects whose responses are best fitted by each of the two costs (*Figure 4e and f*), in comparison with the behaviors of the corresponding best-fitting models (*Figure 4g and h*). This provides a finer understanding of the behavior of subjects than the group average shown in *Figure 2*. For the subjects best fitted by precision-cost models, the proportion of predictions A, $\bar{p}(A)$, when the stimulus generative probability is close to 0.5, is a less steep function of this probability than for the subjects best-fitted by unpredictability-cost models (*Figure 4e and f*, grey lines); furthermore, their sequential effects are larger (as measured by the difference $\bar{p}(A|A) - \bar{p}(A|B)$), and do not depend much on the stimulus generative probability (*Figure 4e and f*, red dots). The corresponding models reproduce the behavioral patterns of the subjects that they best fit (*Figure 4g and h*). Each family of models seems to capture specific behaviors exhibited by the subjects: when fitting the unpredictability-cost models to the responses of the subjects that are best fitted by precision-cost models, and conversely when fitting the precision-cost models to the responses of the subjects that are best fitted by unpredictability-cost models, the models do not reproduce well the subjects' behavioral patterns (*Figure 4i and j*). The precision-cost models, however, seem slightly better than the unpredictability-cost models at capturing the behavior of the subjects that they do not best fit (*Figure 4*, compare panel j to panel f, and panel i to panel e). Substantiating this observation, the examination of the distributions of the models' BICs across subjects shows that when fitting the models onto the subjects that they do not best fit, the precision-cost models fare better than the unpredictability-cost models (see Appendix).

## Beyond the most recent stimulus: patterns of higher-order sequential effects

Notwithstanding the quantitative differences just presented, both families of models yield qualitatively similar attractive sequential effects: the model subjects' predictions are biased towards the preceding stimulus. Does this pattern also apply to the longer history of the stimulus, i.e., do more distant trials also influence the model subjects' predictions? To investigate this hypothesis, we examine the difference between the proportion of predictions A after observing a sequence of length $n$ that starts with A, minus the proportion of predictions A after the same sequence, but starting with B, i.e., $\bar{p}(A|Ax) - \bar{p}(A|Bx)$, where $x$ is a sequence of length $n - 1$, and $Ax$ and $Bx$ denote the same sequence preceded by A and by B. This quantity enables us to isolate the influence of the $n$-to-last stimulus on the current prediction. If the difference is positive, the effect is 'attractive'; if it is negative, the effect is 'repulsive' (in this latter case, the presentation of an A decreases the probability that the subjects predicts A in a later trial, as compared to the presentation of a B); and if the difference is zero there is no sequential effect stemming from the $n$-to-last stimulus. The case $n = 1$ corresponds to the immediately preceding stimulus, whose effect we have shown to be attractive, i.e., $\bar{p}(A|A) - \bar{p}(A|B) > 0$, in the responses both of the best-fitting models and of the subjects (*Figures 2b, 4g and h*).

We investigate the effect of the $n$-to-last stimulus on the behavior of the two families of models, with $n = 1, 2$, and 3. We present here the main results of this investigation; we refer the reader to Methods for a more detailed analysis. With unpredictability-cost models of Markov order $m$, there are non-vanishing sequential effects stemming from the $n$-to-last stimulus only if the Markov order is greater than or equal to the distance from this stimulus to the current trial, i.e., if $m \geq n$. In this case, the sequential effects are attractive (*Figure 5*).

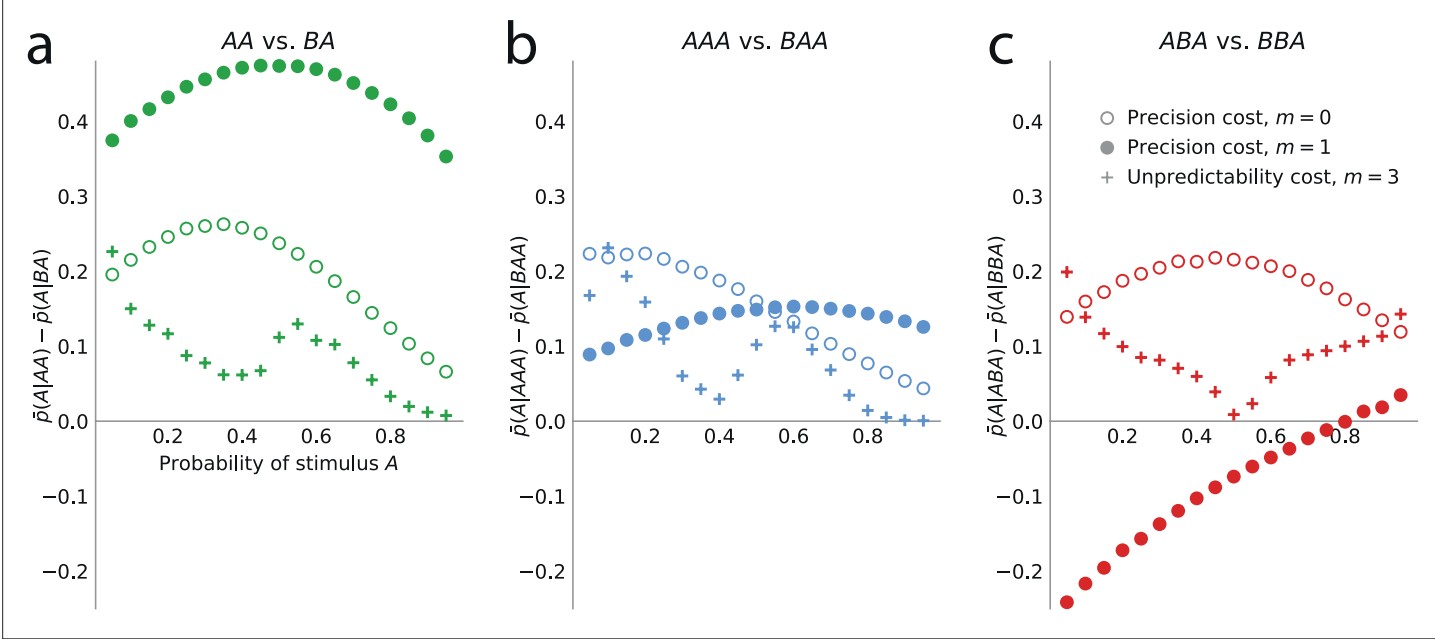

**Figure 5.** Higher-order sequential effects: the precision-cost model of a Markov observer predicts a repulsive effect of the third-to-last stimulus. Sequential effect of the second-to-last (**a**) and third-to-last (**b,c**) stimuli, in the responses of the precision-cost model of a Bernoulli observer ($m = 0$; white circles), of the precision-cost model of a Markov observer with $m = 1$ (filled circles), and of the unpredictability-cost model of a Markov observer with $m = 3$ (crosses). (**a**) Difference between the proportion of prediction A conditional on observing AA, and conditional on observing BA, i.e., $\bar{p}(A|AA) - \bar{p}(A|BA)$, as a function of the stimulus generative probability. With the three models, this difference is positive, indicating an attractive sequential effect of the second-to-last stimulus. (**b**) Difference between the proportion of prediction A conditional on observing AAA, and conditional on observing BAA, i.e., $\bar{p}(A|AAA) - \bar{p}(A|BAA)$. The positive difference indicates an attractive sequential effect of the third-to-last stimulus in this case. (**c**) Difference between the proportion of prediction A conditional on observing ABA, and conditional on observing BBA, i.e., $\bar{p}(A|ABA) - \bar{p}(A|BBA)$. With the precision-cost model of a Markov observer, the negative difference when the stimulus generative probability is lower than 0.8 indicates a repulsive sequential effect of the third-to-last stimulus in this case, while when the probability is greater than 0.8, and with the predictability-cost model of a Bernoulli observer and with the unpredictability-cost model of a Markov observer, the positive difference indicates an attractive sequential effect of the third-to-last stimulus.

With precision-cost models, the $n$-to-last stimuli yield non-vanishing sequential effects regardless of the Markov order, $m$. With $n = 1$, the effect is attractive, i.e., $\bar{p}(A|A) - \bar{p}(A|B) > 0$. With $n = 2$ (second-to-last stimulus), the effect is also attractive, i.e., in the case of the pair of sequences AA and BA, $\bar{p}(A|AA) - \bar{p}(A|BA) > 0$ (**Figure 5a**). By symmetry, the difference is also positive for the other pair of relevant sequences, AB and BB (e.g. we note that $\bar{p}(A|AB) = 1 - \bar{p}(B|AB)$, and that $\bar{p}(B|AB)$ when the probability of A is $p$ is equal to $\bar{p}(A|BA)$ when the probability of A is $1 - p$. We detail in Methods such relations between the proportions of predictions A or B in different situations. These relations result in the symmetries of **Figure 2**, for the sequential effect of the last stimulus, while for higher-order sequential effects they imply that we do not need to show, in **Figure 5**, the effects following all possible past sequences of two or three stimuli, as the ones we do not show are readily derived from the ones we do.)

As for the third-to-last stimulus ($n = 3$), it can be followed by four different sequences of length two, but we only need to examine two of these four, for the reasons just presented. We find that for the precision-cost models, with all the Markov orders we examine (from 0 to 3), the probability of predicting A after observing the sequence AAA is greater than that after observing the sequence BAA, i.e., $\bar{p}(A|AAA) - \bar{p}(A|BAA) > 0$, that is, there is an attractive sequential effect of the third-to-last stimulus if the sequence following it is AA (and, by symmetry, if it is BB; **Figure 5b**). So far, thus, we have found only attractive effects. However, the results are less straightforward when the third-to-last stimulus is followed by the sequence BA. In this case, for a Bernoulli observer ($m = 0$), the effect is also attractive: $\bar{p}(A|ABA) - \bar{p}(A|BBA) > 0$ (**Figure 5c**, white circles). With Markov observers ($m \geq 1$), over a range of stimulus generative probability $p$, the effect is repulsive: $\bar{p}(A|ABA) - \bar{p}(A|BBA) < 0$, that is, the presentation of an A *decreases* the probability that the model subject predicts A three trials later, as

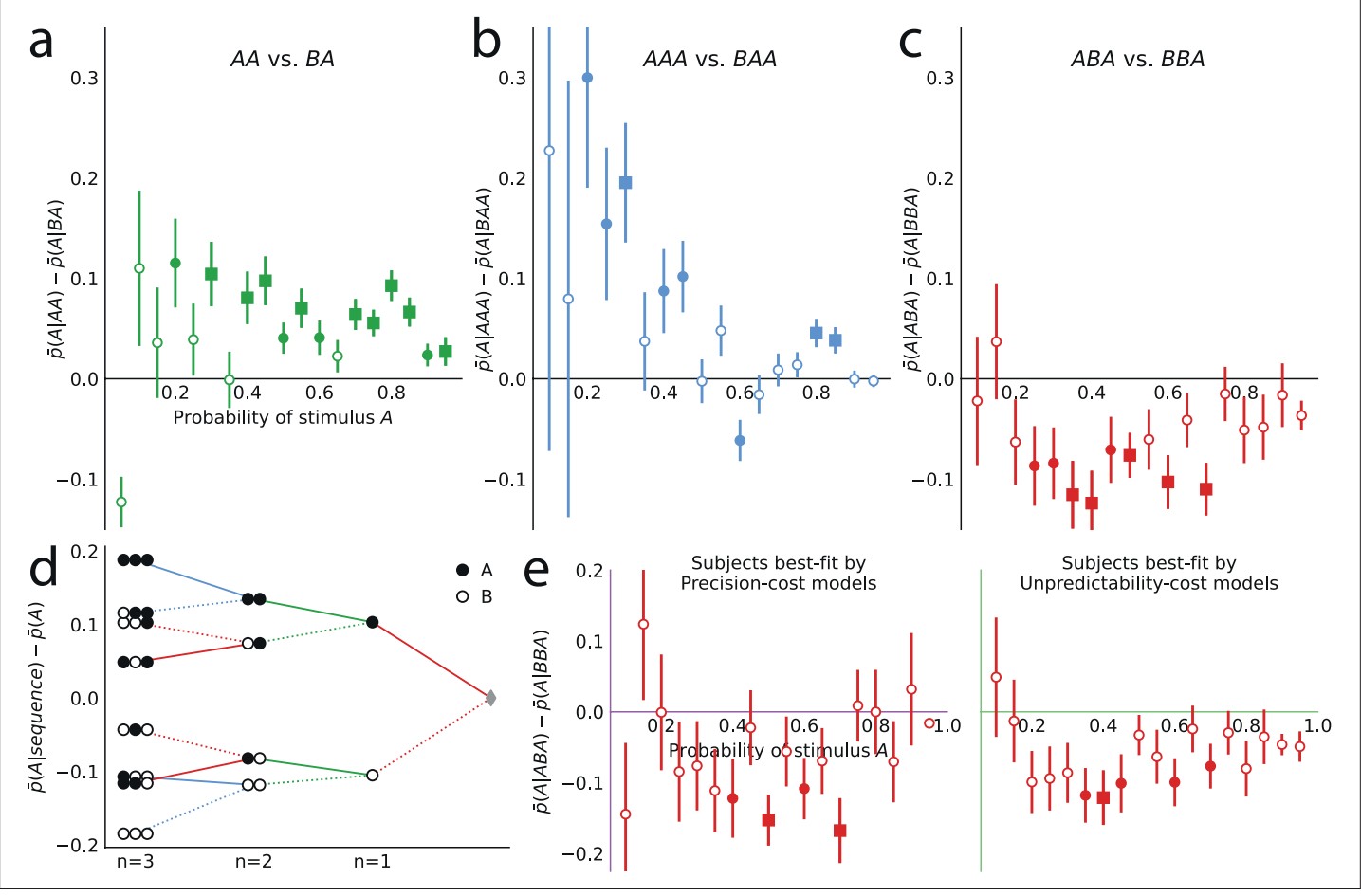

**Figure 6.** Patterns of attractive and repulsive sequential effects in subjects' responses. (**a**) Difference between the proportion of prediction A conditional on observing the sequence AA, and conditional on observing BA, i.e., $\bar{p}(A|AA) - \bar{p}(A|BA)$, as a function of the stimulus generative probability. This difference is in most cases positive, indicating an attractive sequential effect of the second-to-last stimulus. (**b**) Difference between the proportion of prediction A conditional on observing AAA, and conditional on observing BAA, i.e., $\bar{p}(A|AAA) - \bar{p}(A|BAA)$. This difference is positive in most cases, indicating an attractive sequential effect of the third-to-last stimulus. (**c**) Difference between the proportion of prediction A conditional on observing ABA, and conditional on observing BBA, i.e., $\bar{p}(A|ABA) - \bar{p}(A|BBA)$. This difference is negative in most cases, indicating a repulsive sequential effect of the third-to-last stimulus. (**d**) Differences, averaged over all tested stimulus generative probabilities, between the proportion of predictions A conditional on sequences of up to three past observations, minus the unconditional proportion. The proportion conditional on a sequence is an average of the two proportions conditional on the same sequence preceded by another, 'older' observation, A or B, resulting in a binary-tree structure in this representation. If this additional past observation is A (respectively, B), we connect the two sequences with a solid line (respectively, a dotted line). In most cases, conditioning on an additional A increases the proportion of predictions A (in comparison to conditioning on an additional B), indicating an attractive sequential effect, except when the additional observation precedes the sequence BA (or its symmetric AB), in which cases repulsive sequential effects are observed (dotted line 'above' solid line). (**e**) Same as (**c**), with subjects split in two groups: the subjects best-fitted by precision-cost models (left) and the subjects best-fitted by unpredictability-cost models (right). In panels **a-c** and **e**, the filled circles indicate that the p-value of the Fisher exact test is below 0.05, and the filled squares indicate that the p-value with Bonferroni-Holm-Šidák correction is below 0.05. Bars are twice the square root of the sum of the two squared standard errors of the means (for each point, total n: a: 178 to 3584, b: 37 to 3394, c: 171 to 1868, e: 63 to 1184). In all panels, the responses of all the subjects are pooled together.

compared to the presentation of a B (*Figure 5c*, filled circles). The occurrence of the repulsive effect in this particular case is a distinctive trait of the precision-cost models of Markov observers ($m \geq 1$); we do not obtain any repulsive effect with any of the unpredictability-cost models, nor with the precision-cost model of a Bernoulli observer ($m = 0$).

## Subjects' predictions exhibit higher-order repulsive effects

We now examine the sequential effects in subjects' responses, beyond the attractive effect of the preceding stimulus ($n = 1$; discussed above). With $n = 2$ (second-to-last stimulus), for the majority

of the 19 stimulus generative probabilities $p$, we find attractive sequential effects: the difference $\bar{p}(A|AA) - \bar{p}(A|BA)$ is significantly positive (**Figure 6a**; p-values <0.01 for 11 stimulus generative probabilities, <0.05 for 13 probabilities; subjects pooled). With $n = 3$ (third-to-last stimulus), we also find significant attractive sequential effects in subjects' responses for some of the stimulus generative probabilities, when the third-to-last stimulus is followed by the sequence AA (**Figure 6b**; p-values <0.01 for four probabilities, <0.05 for seven probabilities). When it is instead followed by the sequence BA, we find that for eight stimulus generative probabilities, all between 0.25 and 0.75, there is a significant repulsive sequential effect: $\bar{p}(A|ABA) - \bar{p}(A|BBA) < 0$ (p-values <0.01 for six probabilities, <0.05 for eight probabilities; subjects pooled). Thus, in these cases, the occurrence of A as the third-to-last stimulus increases (in comparison with the occurrence of a B) the proportion of the *opposite* prediction, B. For the remaining stimulus generative probabilities, this difference is in most cases also negative although not significantly different from zero (**Figure 6c**). (An across-subjects analysis yields similar results; see Supplementary Materials.) **Figure 6d** summarizes subjects' sequential effects, and exhibits the attractive and repulsive sequential effects in their responses (compare solid and dotted lines). (In this tree-like representation, we show averages across the stimulus generative probabilities; a figure with the individual 'trees' for each probability is provided in the Appendix.)

The repulsive sequential effect of the third-to-last stimulus in subjects' predictions only occurs when the third-to-last stimulus is A followed by the sequence BA. It is also only in this case that the repulsive effect appears with the precision-cost models of a Markov observer (while it never appears with the unpredictability-cost models). This qualitative difference suggests that the precision-cost models offer a better account of sequential effects in subjects. However, model-fitting onto the overall behavior presented above showed that a fraction of the subjects is better fitted by the unpredictability-cost models. We investigate, thus, the presence of a repulsive effect in the predictions of the subjects best fitted by the precision-cost models, and of those best fitted by the unpredictability-cost models. For the subjects best fitted by the precision-cost models, we find (expectedly) that there is a significant repulsive sequential effect of the third-to-last stimulus ($\bar{p}(A|ABA) - \bar{p}(A|BBA) < 0$; p-values <0.01 for two probabilities, <0.05 for four probabilities; subjects pooled; **Figure 6e**, left panel). For the subjects best fitted by the unpredictability-cost models (a family of model that does not predict any repulsive sequential effects), we also find, perhaps surprisingly, a significant repulsive effect of the third-to-last stimulus (p-values <0.01 for three probabilities, <0.05 for five probabilities; subjects pooled), which demonstrates the robustness of this effect (**Figure 6e**, right panel). Thus, in spite of the results of the model-selection procedure, some sequential effects in subjects' predictions support only one of the two families of model. Regardless of the model that best fits their overall predictions, the behavior of the subjects is consistent only with the precision-cost family of models with Markov order equal to or greater than 1, that is, with a model of inference of conditional probabilities hampered by a cognitive cost weighing on the precision of belief distributions.

## Discussion

We investigated the hypothesis that sequential effects in human predictions result from cognitive constraints hindering the inference process carried out by the brain. We devised a framework of constrained inference, in which the model subject bears a cognitive cost when updating its belief distribution upon the arrival of new evidence: the larger the cost, the more the subject's posterior differs from the Bayesian posterior. The models we derive from this framework make specific predictions. First, the proportion of forced-choice predictions for a given stimulus should increase with the stimulus generative probability. Second, most of those models predict sequential effects: predictions also depend on the recent stimulus history. Models with different types of cognitive cost resulted in different patterns of attractive and repulsive effects of the past few stimuli on predictions. To compare the predictions of constrained inference with human behavior, we asked subjects to predict each next outcome in sequences of binary stimuli. We manipulated the stimulus generative probability in blocks of trials, exploring exhaustively the probability range from 0.05 to 0.95 by increments of 0.05. We found that subjects' predictions depend on both the stimulus generative probability and the recent stimulus history. Sequential effects exhibited both attractive and repulsive components which were modulated by the stimulus generative probability. This behavior was qualitatively accounted for by a model of constrained inference in which the subject infers the transition probabilities underlying the sequences of stimuli and bears a cost that increases with the precision of the posterior distributions.

Our study proposes a novel theoretical account of sequential effects in terms of optimal inference under cognitive constraints and it uncovers the richness of human behavior over a wide range of stimulus generative probabilities.

The notion that human decisions can be understood as resulting from a constrained optimization has gained traction across several fields, including neuroscience, cognitive science, and economics. In neuroscience, a voluminous literature that started with *Attneave, 1954* and *Barlow, 1961* investigates the idea that perception maximizes the transmission of information, under the constraint of costly and limited neural resources (*Laughlin, 1981*; *Laughlin et al., 1998*; *Simoncelli and Olshausen, 2001*); related theories of 'efficient coding' account for the bias and the variability of perception (*Ganguli and Simoncelli, 2016*; *Wei and Stocker, 2015*; *Wei and Stocker, 2017*; *Prat-Carrabin and Woodford, 2021c*). In cognitive science and economics, 'bounded rationality' is a precursory concept introduced in the 1950s by Herbert Simon, who defines it as "rational choice that takes into account the cognitive limitations of the decision maker — limitations of both knowledge and computational capacity" (*Simon, 1997*). For Gigerenzer, these limitations promote the use of heuristics, which are 'fast and frugal' ways of reasoning, leading to biases and errors in humans and other animals (*Gigerenzer and Goldstein, 1996*; *Gigerenzer and Selten, 2002*). A range of more recent approaches can be understood as attempts to specify formally the limitations in question, and the resulting trade-off. The 'resource-rational analysis' paradigm aims at a unified theoretical account that reconciles principles of rationality with realistic constraints about the resources available to the brain when it is carrying out computations (*Griffiths et al., 2015*). In this approach, biases result from the constraints on resources, rather than from 'simple heuristics' (see *Lieder and Griffiths, 2019* for an extensive review). For instance, in economics, theories of 'rational inattention' propose that economic agents optimally allocate resources (a limited amount of attention) to make decisions, thereby proposing new accounts of empirical findings in the economic literature (*Sims, 2003*; *Woodford, 2009*; *Caplin et al., 2019*; *Gabaix, 2017*; *Azeredo da Silveira and Woodford, 2019*; *Azeredo da Silveira et al., 2020*).

Our study puts forward a 'resource-rational' account of sequential effects. Traditional accounts since the 1960s attribute these effects to a belief in sequential dependencies between successive outcomes (*Edwards, 1961*; *Matthews and Sanders, 1984*) (potentially 'acquired through life experience' *Ayton and Fischer, 2004*), and more generally to the incorrect models that people assume about the processes generating sequences of events (see *Oskarsson et al., 2009* for a review; similar rationales have been proposed to account for suboptimal behavior in other contexts, for example in exploration-exploitation tasks *Navarro et al., 2016*). This traditional account was formalized, in particular, by models in which subjects carry out a statistical inference about the sequence of stimuli presented to them, and this inference assumes that the parameters underlying the generating process are subject to changes (*Yu and Cohen, 2008*; *Wilder et al., 2009*; *Zhang et al., 2014*; *Meyniel et al., 2016*). In these models, sequential effects are thus understood as resulting from a rational adaptation to a changing world. Human subjects indeed dynamically adapt their learning rate when the environment changes (*Payzan-LeNestour et al., 2013*; *Meyniel and Dehaene, 2017*; *Nassar et al., 2010*), and they can even adapt their inference to the statistics of these changes (*Behrens et al., 2007*; *Prat-Carrabin et al., 2021b*). However, in our task and in many previous studies in which sequential effects have been reported, the underlying statistics are in fact not changing across trials. The models just mentioned thus leave unexplained why subjects' behavior, in these tasks, is not rationally adapted to the unchanging statistics of the stimulus.

What underpins our main hypothesis is a different kind of rational adaptation: one, instead, to the 'cognitive limitations of the decision maker', which we assume hinder the inference carried out by the brain. We show that rational models of inference under a cost yield rich patterns of sequential effects. When the cost varies with the precision of the posterior (measured here by the negative of its entropy, *Equation 3*), the resulting optimal posterior is proportional to the product of the prior and the likelihood, each raised to an exponent $1/(\lambda + 1)$ (*Equation 4*). Many previous studies on biased belief updating have proposed models that adopt the same form except for the different exponents applied to the prior and to the likelihood (*Grether, 1980*; *Matsumori et al., 2018*; *Benjamin, 2019*). Here, with the precision cost, both quantities are raised to the same exponent and we note that in this case the inference of the subject amounts to an exponentially decaying count of the patterns observed in the sequence of stimuli, which is sometimes called 'leaky integration' in the literature (*Yu and Cohen, 2008*; *Wilder et al., 2009*; *Jones et al., 2013*; *Meyniel et al., 2016*). The models mentioned above,

that posit a belief in changing statistics, indeed are well approximated by models of leaky integration (*Yu and Cohen, 2008*; *Meyniel et al., 2016*), which shows that the exponential discount can have different origins. *Meyniel et al., 2016* show that the precision-cost, Markov-observer model with $m = 1$ (named 'local transition probability model' in this study) accounts for a range of other findings, in addition to sequential effects, such as biases in the perception of randomness and patterns in the surprise signals recorded through EEG and fMRI. Here we reinterpret these effects as resulting from an optimal inference subject to a cost, rather than from a suboptimal erroneous belief in the dynamics of the stimulus' statistics. In our modeling approach, the minimization of a loss function (*Equation 1*) formalizes a trade-off between the distance to optimality of the inference, and the cognitive constraints under which it is carried out. We stress that our proposal is not that the brain actively solves this optimization problem online, but instead that it is endowed with an inference algorithm (whose origin remains to be elucidated) which is effectively a solution to the constrained optimization problem.

By grounding the sequential effects in the optimal solution to a problem of constrained optimization, our approach opens avenues for exploring the origins of sequential effects, in the form of hypotheses about the nature of the constraint that hinders the inference carried out by the brain. With the precision cost, more precise posterior distributions are assumed to take a larger cognitive toll. The intuitive assumption that it is costly to be precise finds a more concrete realization in neural models of inference with probabilistic population codes: in these models, the precision of the posterior is proportional to the average activity of the population of neurons and to the number of neurons (*Ma et al., 2006*; *Seung and Sompolinsky, 1993*). More neural activity and more neurons arguably come with a metabolic cost, and thus more precise posteriors are more costly in these models. Imprecisions in computations, moreover, was shown to successfully account for decision variability and adaptive behavior in volatile environments (*Findling et al., 2019*; *Findling et al., 2021*).

The unpredictability cost, which we introduce, yields models that also exhibit sequential effects (for Markov observers), and that fit several subjects better than the precision-cost models. The unpredictability cost relies on a different hypothesis: that the cost of representing a distribution over different possible states of the world (here, different possible values of $q$) resides in the difficulty of representing these states. This could be the case, for instance, under the hypothesis that the brain runs stochastic simulations of the implied environments, as proposed in models of 'intuitive physics' (*Battaglia et al., 2013*) and in Kahneman and Tversky's 'simulation heuristics' (*Kahneman et al., 1982*). More entropic environments imply more possible scenarios to simulate, giving rise, under this assumption, to higher costs. A different literature explores the hypothesis that the brain carries out a mental compression of sequences (*Simon, 1972*; *Chekaf et al., 2016*; *Planton et al., 2021*); entropy in this context is a measure of the degree of compressibility of a sequence (*Planton et al., 2021*), and thus, presumably, of its implied cost. As a result, the brain may prefer predictable environments over unpredictable ones. Human subjects exhibit a preference for predictive information indeed (*Ogawa and Watanabe, 2011*; *Trapp et al., 2015*), while unpredictable stimuli have been shown not only to increase anxiety-like behavior (*Herry et al., 2007*), but also to induce more neural activity (*Herry et al., 2007*; *den Ouden et al., 2009*; *Alink et al., 2010*) — a presumably costly increase, which may result from the encoding of larger prediction errors (*Herry et al., 2007*; *Schultz and Dickinson, 2000*).

We note that both costs (precision and unpredictability) can predict sequential effects, even though neither carries *ex ante* an explicit assumption that presupposes the existence of sequential effects. They both reproduce the attractive recency effect of the last stimulus exhibited by the subjects. They make quantitatively different predictions (*Figure 4*); we also find this diversity of behaviors in subjects.

The precision cost, as mentioned above, yields leaky-integration models which can be summarized by a simple algorithm in which the observed patterns are counted with an exponential decay. The psychology and neuroscience literature proposes many similar 'leaky integrators' or 'leaky accumulators' models (*Smith, 1995*; *Roe et al., 2001*; *Usher and McClelland, 2001*; *Cook and Maunsell, 2002*; *Wang, 2002*; *Sugrue et al., 2004*; *Bogacz et al., 2006*; *Kiani et al., 2008*; *Yu and Cohen, 2008*; *Gao et al., 2011*; *Tsetsos et al., 2012*; *Ossmy et al., 2013*; *Meyniel et al., 2016*). In connectionist models of decision-making, for instance, decision units in abstract network models have activity levels that accumulate evidence received from input units, and which decay to zero in the absence of input (*Roe et al., 2001*; *Usher and McClelland, 2001*; *Wang, 2002*; *Bogacz et al., 2006*; *Tsetsos et al., 2012*). In other instances, perceptual evidence (*Kiani et al., 2008*; *Gao et al., 2011*; *Ossmy*

*et al., 2013*) or counts of events (*Sugrue et al., 2004*; *Yu and Cohen, 2008*; *Meyniel et al., 2016*) are accumulated through an exponential temporal filter. In our approach, leaky integration is not an assumption about the mechanisms underpinning some cognitive process: instead, we find that it is an optimal strategy in the face of a cognitive cost weighing on the precision of beliefs. Although it is less clear whether the unpredictability-cost models lend themselves to a similar algorithmic simplification, they consist in a distortion of Bayesian inference, for which various neural-network models have been proposed (*Deneve et al., 2001*; *Ma et al., 2008*; *Ganguli and Simoncelli, 2014*; *Echeveste et al., 2020*).

Turning to the experimental results, we note that in spite of the rich literature on sequential effects, the majority of studies have focused on equiprobable Bernoulli environments, in which the two possible stimuli both had a probability equal to 0.5, as in tosses of a fair coin (*Soetens et al., 1985*; *Cho et al., 2002*; *Yu and Cohen, 2008*; *Wilder et al., 2009*; *Jones et al., 2013*; *Zhang et al., 2014*; *Ayton and Fischer, 2004*; *Gökaydin and Ejova, 2017*). In environments of this kind, the two stimuli play symmetric roles and all sequences of a given length are equally probable. In contrast, in biased environments one of the two possible stimuli is more probable than the other. Although much less studied, this situation breaks the regularities of equiprobable environments and is arguably very frequent in real life. In our experiment, we explore stimulus generative probabilities from 0.05 to 0.95, thus allowing to investigate the behavior of subjects in a wide spectrum of Bernoulli environments: from these with 'extreme' probabilities (e.g. $p = 0.95$) to these only slightly different from the equiprobable case (e.g. $p = 0.55$) to the equiprobable case itself ($p = 0.5$). The subjects are sensitive to the imbalance of the non-equiprobable cases: while they predict A in half the trials of the equiprobable case, a probability of just $p = 0.55$ suffices to prompt the subjects to predict A in about in 60% of trials, a significant difference ($\bar{p}(A) = 0.602$; sem: 0.008; p-value of t-test against null hypothesis that $\bar{p}(A) = 0.5$: 1.7e-11; subjects pooled).

The well-known 'probability matching' hypothesis (*Herrnstein, 1961*; *Vulkan, 2000*; *Gaissmaier and Schooler, 2008*) suggests that the proportion of predictions A matches the stimulus generative probability: $\bar{p}(A) = p$. This hypothesis is not supported by our data. We find that in the non-equiprobable conditions these two quantities are significantly different (all p-values <1e-11, when $p \neq 0.5$). More precisely, we find that the proportion of prediction A is more extreme than the stimulus generative probability (i.e. $\bar{p}(A) > p$ when $p > 0.5$, and $\bar{p}(A) < p$ when $p < 0.5$; *Figure 2a*). This result is consistent with the observations made by *Edwards, 1961*; *Edwards, 1956* and with the conclusions of a more recent review (*Vulkan, 2000*).

In addition to varying with the stimulus generative probability, the subjects' predictions depend on the recent history of stimuli. Recency effects are common in the psychology literature; they were reported from memory (*Ebbinghaus et al., 1913*) to causal learning (*Collins and Shanks, 2002*) to inference (*Shanteau, 1972*; *Hogarth and Einhorn, 1992*; *Benjamin, 2019*). Recency effects, in many studies, are obtained in the context of reaction tasks, in which subjects must identify a stimulus and quickly provide a response (*Hyman, 1953*; *Bertelson, 1965*; *Kornblum, 1967*; *Soetens et al., 1985*; *Cho et al., 2002*; *Yu and Cohen, 2008*; *Wilder et al., 2009*; *Jones et al., 2013*; *Zhang et al., 2014*). Although our task is of a different kind (subjects must predict the next stimulus), we find some evidence of recency effects in the response times of subjects: after observing the less frequent of the two stimuli (when $p \neq 0$), subjects seem slower at providing a response (see Appendix). In prediction tasks (like ours), both attractive recency effects, also called 'hot-hand fallacy', and repulsive recency effects, also called 'gambler's fallacy', have been reported (*Jarvik, 1951*; *Edwards, 1961*; *Ayton and Fischer, 2004*; *Burns and Corpus, 2004*; *Croson and Sundali, 2005*; *Oskarsson et al., 2009*). The observation of both effects within the same experiment has been reported in a visual identification task (*Chopin and Mamassian, 2012*) and in risky choices ('wavy recency effect' *Plonsky et al., 2015*; *Plonsky and Erev, 2017*). As to the heterogeneity of these results, several explanations have been proposed; two important factors seem to be the perceived degree of randomness of the predicted variable and whether it relates to human performance (*Ayton and Fischer, 2004*; *Burns and Corpus, 2004*; *Croson and Sundali, 2005*; *Oskarsson et al., 2009*). In any event, most studies focus exclusively on the influence of 'runs' of identical outcomes on the upcoming prediction, for example, in our task, on whether three As in a row increases the proportion of predictions A. With this analysis, Edwards (*Edwards, 1961*) in a task similar to ours concluded to an attractive recency effect (which he called 'probability following'). Although our results are consistent with this observation (in our data

three As in a row do increase the proportion of predictions A), we provide a more detailed picture of the influence of each stimulus preceding the prediction, whether it is in a 'run' of identical stimuli or not, which allows us to exhibit the non-trivial finer structure of the recency effects that is often overlooked.

Up to two stimuli in the past, the recency effect is attractive: observing A at trial $t-2$ or at trial $t-1$ induces, all else being equal, a higher proportion of predictions A at trial $t$ (in comparison to observing B; **Figures 2 and 6a**). The influence of the third-to-last stimulus is more intricate: it can yield either an attractive or a repulsive effect, depending on the second-to-last and the last stimuli. For a majority of probability parameters, $p$, while an A followed by the sequence AA has an attractive effect (i.e. $p(A|AAA) > p(A|BAA)$), an A followed by the sequence BA has a repulsive effect (i.e. $p(A|ABA) < p(A|BBA)$; **Figure 6b and c**). How can this reversal be intuited? Only one of our models, the precision-cost model with a Markov order 1 ($m=1$), reproduces this behavior; we show how it provides an interpretation for this result. From the update equation of this model (**Equation 4**), it is straightforward to show that the posterior of the model subject (a Dirichlet distribution of order 4) is determined by four quantities, which are exponentially-decaying counts of the four two-long patterns observed in the sequence of stimuli: BB, BA, AB, and AA. The higher the count of a pattern, the more likely the model subject deems this pattern to happen again. In the equiprobable case ($p=0.5$), after observing the sequence AAA, the count of AA is higher than after observing BAA, thus the model subject believes that AA is more probable, and accordingly predicts A more frequently, i.e., $p(A|AAA) > p(A|BAA)$. As for the sequences ABA and BBA, both result in the same count of AA, but the former results in a higher count of AB — in other words, the short sequence ABA suggests that A is usually followed by B, but the sequence BBA does not — and thus the model subject predicts more frequently B, i.e., less frequently A ($p(A|ABA) < p(A|BBA)$).

In short, the ability of the precision-cost model of a Markov observer to capture the repulsive effect found in behavioral data suggests that human subjects extrapolate the local statistical properties of the presented sequence of stimuli in order to make predictions, and that they pay attention not only to the 'base rate' — the marginal probability of observing A, unconditional on the recent history — as a Bernoulli observer would do, but also to the statistics of more complex patterns, including the repetitions and the alternations, thus capturing the transition probabilities between consecutive observations. **Wilder et al., 2009**, **Jones et al., 2013**, and **Meyniel et al., 2016** similarly argue that sequential effects result from an imperfect inference of the base rate and of the frequency of repetitions and alternations. **Dehaene et al., 2015** argue that the knowledge of transition probabilities is a central mechanism in the brain's processing of sequences (e.g. in language comprehension), and infants as young as 5 months were shown to be able to track both base rates and transition probabilities (see **Saffran and Kirkham, 2018** for a review). Learning of transition probabilities has also been observed in rhesus monkeys (**Meyer and Olson, 2011**).

The deviations from perfect inference, in the precision-cost model, originate in the constraints faced by the brain when performing computation with probability distributions. In spite of the success of the Bayesian framework, we note that human performance in various inference tasks is often suboptimal (**Nassar et al., 2010**; **Hu et al., 2013**; **Acerbi et al., 2014**; **Prat-Carrabin et al., 2021b**; **Prat-Carrabin and Woodford, 2022**). Our approach suggests that the deviations from optimality in these tasks may be explained by the cognitive constraints at play in the inference carried out by humans.

Other studies have considered the hypothesis that suboptimal behavior in inference tasks results from cognitive constraints. **Kominers et al., 2016** consider a model in which Bayesian inference comes with a fixed cost; the observer can choose to forgo updating her belief, so as to avoid the cost. In some cases, the model predicts 'permanently cycling beliefs' that do not converge; but in general the model predicts that subjects will choose not to react to new evidence that is unsurprising under the current belief. The significant sequential effects we find in our subjects' responses, however, seem to indicate that they are sensitive to both unsurprising (e.g. outcome A when p>0.5) and surprising (outcome B when p>0.5) observations, at least across the values of the stimulus generative probability that we test (**Figure 2**). **Graeber, 2020** considers costly information processing as an account of subjects' neglect of confounding variables in an inference task, but concludes instead that the suboptimal behavior of subjects results from their misunderstanding of the information structure in the task. A model close to ours is the one proposed in **Azeredo da Silveira and Woodford, 2019** and **Azeredo da Silveira et al., 2020**, in which an information-theoretic cost limits the memory of an

otherwise optimal and Bayesian decision-maker, resulting, here also, in beliefs that fluctuate and do not converge, and in an overweighting, in decisions, of the recent evidence.

Taking a different approach, **Dasgupta et al., 2020** implement a neural network that learns to approximate Bayesian posteriors. Possible approximate posteriors are constrained not only by the structure of the network, but also by the fact that the same network is used to address a series of different inference problems. Thus the network's parameters must be 'shared' across problems, which is meant to capture the brain's limited computational resources. Although this constraint differs from the ones we consider, we note that in this study the distance function (which the approximation aims to minimize) is the same as in our models, namely, the Kullback-Leibler divergence from the optimal posterior to the approximate posterior, $D_{KL}(\hat{P}\|P)$. Minimizing this divergence (under a cost) allows the model subject to obtain a posterior as close as possible (at least by this measure) to the optimal posterior given the most recent stimulus and the subject's belief prior to observing the stimulus, which in turn enables the subject to perform reasonably well in the task.

In principle, rewarding subjects with a higher payoff when they make a correct prediction would change the optimal trade-off (between the distance to the optimal posterior and the cognitive costs) formalized in **Equation 1**, resulting in 'better' posteriors (closer to the Bayesian posterior), and thus to higher performance in the task. At the same time, incentivization is known to influence, also in the direction of higher performance, the extent to which choice behavior is close to probability matching (**Vulkan, 2000**). The interesting question of the respective sensitivities of the subjects' inference process and of their response-selection strategy in response to different levels of incentives is beyond the scope of this study, in which we have focussed on the sensitivity of behavior to different stimulus generative probabilities.

In any case, the approach of minimizing the Kullback-Leibler divergence from the optimal posterior to the approximate posterior is widely used in the machine learning literature, and forms the basis of the 'variational' family of approximate-inference techniques (**Bishop, 2006**). These techniques have inspired various cognitive models (**Sanborn, 2017**; **Gallistel and Latham, 2022**; **Aridor and Woodford, 2023**); alternatively, a bound on the divergence, known as the 'evidence bound', or, in neuroscience, as the negative of the 'free energy', is maximized (**Moustafa, 2017**; **Friston et al., 2006**; **Friston, 2009**). (We note that the 'opposite' divergence, $D_{KL}(P\|\hat{P})$, is minimized in a different machine-learning technique, 'expectation propagation' (**Bishop, 2006**), and in the cognitive model of causal reasoning of **Icard and Goodman, 2015**.) In these techniques, the approximate posterior is chosen within a convenient family of tractable, parameterized distributions; other distributions are precluded. This can be understood, in our framework, as positing a cost $C(\hat{P})$ that is infinite for most distributions, but zero for the distributions that belong to some arbitrary family (**Prat-Carrabin et al., 2021a**). The precision cost and the unpredictability cost, in comparison, are 'smooth', and allow for any distribution, but they penalize, respectively, more precise belief distributions, and belief distributions that imply more unpredictable environments. Our study shows that inference, when subject to either of these costs, yields an attractive sequential effect of the most recent observation; and with a precision cost weighing on the inference of transition probabilities (i.e., $m = 1$), the model predicts the subtle pattern of attractive and repulsive sequential effects that we find in subjects' responses.

## Methods
### Task and subjects

The computer-based task was programmed using the Python library PsychoPy (**Peirce, 2008**). The experiment comprised ten blocks of trials, which differed by the stimulus generative probability, p, used in all the trials of each block. The probability p was chosen randomly among the ten values ranging from 0.50 to 0.95 by increments of 0.05, excluding the values already chosen; and with probability 1/2 the stimulus generative probability $1 - p$ was used instead. Each block started with 200 passive trials, in which the subject was only asked to look at the 200 stimuli sampled with the block's probability and successively presented. No action from the subject was required for these passive trials. The subject was then asked to predict, in each of 200 trials, the next location of the stimulus. Subjects provided their responses by a keypress. The task was presented as a game to the subjects: the stimulus was a lightning symbol, and predicting correctly whether the lightning would strike the left or the right rod resulted in the electrical energy of the lightning being collected in a battery

(*Figure 1*). A gauge below the battery indicated the amount of energy accumulated in the current block of trials (*Figure 1a*). Twenty subjects (7 women, 13 men; age: 18–41, mean 25.5, standard deviation 6.2) participated in the experiment. All subjects completed the ten blocks of trials, except one subject who did not finish the experiment and was excluded from the analysis. The study was approved by the ethics committee Île de France VII (CPP 08–021). Participants gave their written consent prior to participating. The number of blocks of trials and the number of trials per block were chosen as a trade-off between maximizing the statistical power of the study, scanning the values of the generative probability parameter from 0.05 to 0.95 with a satisfying resolution, and maintaining the duration of the experiment under a reasonable length of time. The number of subjects was chosen consistently with similar studies and so as to capture individual variability. Throughout the study, we conduct Student's t-tests when comparing the subjects' proportion of predictions A to a given value (e.g. 0.5). When comparing two proportions of predictions A obtained under different conditions (e.g. depending on whether the preceding stimulus is A or B), we accordingly conduct Fisher exact tests. The trials in which subjects failed to respond within the limit of 1 s were not included in the analysis. They represented 1.27% of the trials, on average (across subjects); and for 95% of the subjects these trials represented less than 2.5% of the trials.

## Sequential effects of the models

We run simulations of the eight models and look at the predictions they yield. To reproduce the conditions faced by the subjects, which included 200 passive trials, we start each simulation by showing to the model subject 200 randomly sampled stimuli (without collecting predictions at this stage). We then show an additional 200 samples, and obtain a prediction from the model subject after each sample. The sequential effects of the most recent stimulus, with the different models, are shown in *Figure 7*. With the precision-cost models, the posterior distribution of the model subject does not converge, but fluctuates instead with the recent history of the stimulus. This results in attractive sequential effects (*Figure 7a*), including for the Bernoulli observer, who assumes that the probability of A does not depend on the most recent stimulus. With the unpredictability-cost models, the posterior of the model subject does converge. With Markov observers, it converges toward a parameter vector $q$ that implies that the probability of observing A depends on the most recent stimulus, resulting in the presence of sequential effects of the most recent stimulus (*Figure 7b*, second to fourth row). With a Bernoulli observer, the posterior of the model subject converges toward a value of the stimulus generative probability that does not depend on the stimulus history. As more evidence is accumulated, the posterior narrows around this value but not without some fluctuations that depend on the sequence of stimuli presented. In consequence the model subject's estimate of the stimulus generative probability is also subject to fluctuations, and depends on the history of stimuli (including the most recent stimulus), although the width of the fluctuations tend to zero as more stimuli are observed. After the 200 stimuli of the passive trials, the sequential effects of the most recent stimulus resulting from this transient regime appear small in comparison to the sequential effects obtained with the other models (*Figure 7b*, first row). The *Figure 7* also shows the behaviors of the models when augmented with a propensity to repeat the preceding response: we comment on these in the section dedicated to these models, below.

Turning to higher-order sequential effects, we look at the influence on predictions of the second- and third-to-last stimulus (*Figure 8*). As mentioned, only precision-cost models of Markov observers yield repulsive sequential effects, and these occur only when the third-to-last-stimulus is followed by BA. They do not occur with the second-to-last stimulus, nor with the third-to-last-stimulus when it is followed by AA (*Figure 8a*); and they do not occur in any case with the unpredictability-cost models (*Figure 8b*).

## Derivation of the approximate posteriors

We derive the solution to the constrained optimization problem, in the general case of a 'hybrid' model subject who bears both a precision cost, with weight $\lambda_p$, and an unpredictability cost, with weight $\lambda_u$. Thus the subject minimizes the loss function

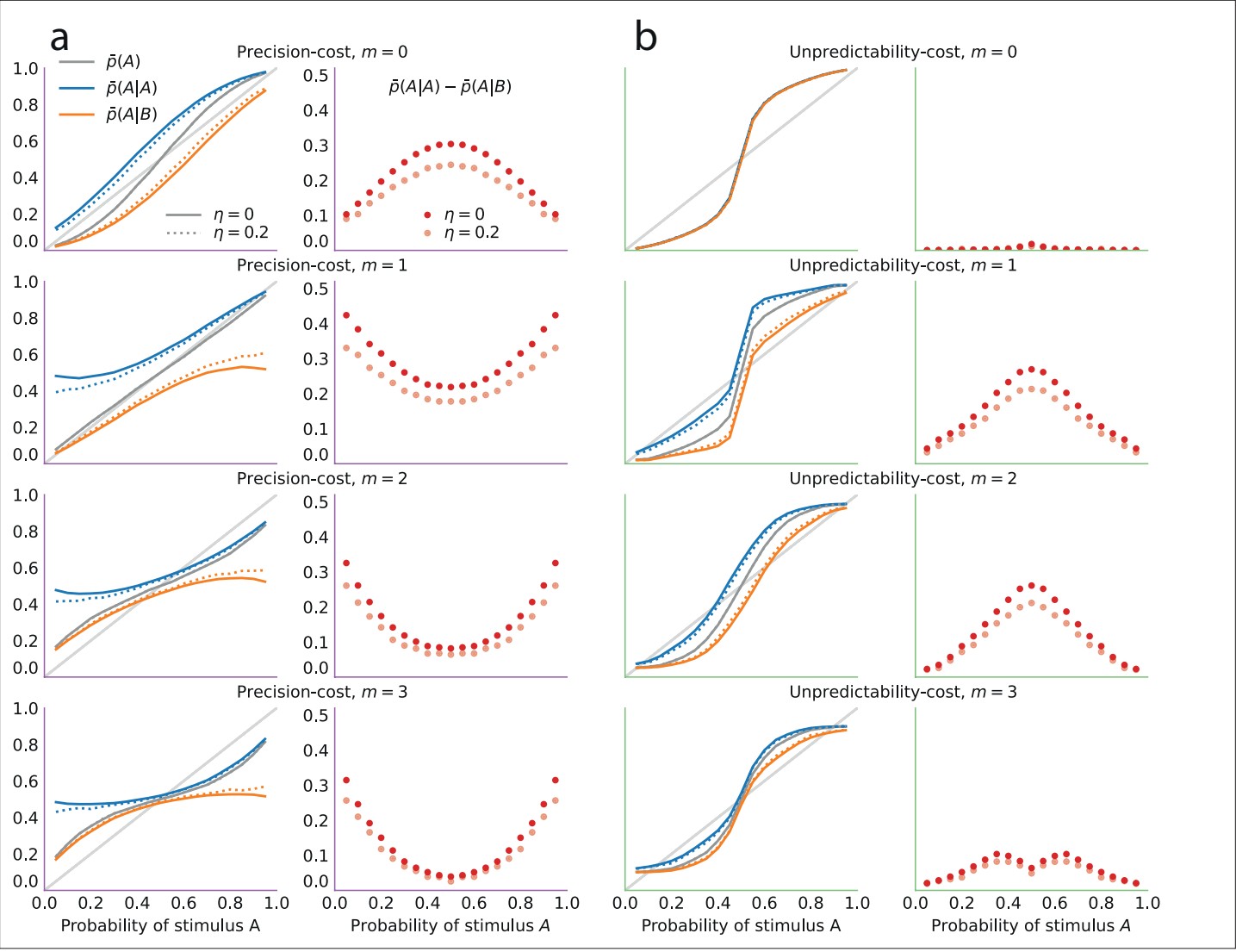

**Figure 7.** Sequential effects of the most recent stimulus in precision-cost and unpredictability-cost models. (**a**) Precision-cost models. (**b**) Unpredictability-cost models. *First row*: Bernoulli observers (m = 0). *Second to fourth rows*: Markov observers (m = 1, 2, and 3). *First column (each panel)*: proportion of predictions A in the models' responses as a function of the stimulus generative probability, conditional on the preceding observation being A (blue line) or B (orange line), and unconditional (grey line); with repetition propensity ($\eta = 0.2$, dotted lines), and without (solid lines). *Second column (each panel)*: difference between the proportion of predictions A conditional on the preceding observation being A, and the same proportion conditional on a B; with repetition propensity (dotted lines), and without (solid lines). A positive difference indicates an attractive sequential effect of the most recent stimulus.

$$
\begin{aligned}
L(\hat{P}_{t+1}) = &\int \hat{P}_{t+1}(q) \ln \frac{\hat{P}_{t+1}(q)}{P_{t+1}(q)} dq \\
&+ \lambda_u \int H(X; q) \hat{P}_{t+1}(q) dq \\
&+ \lambda_p \int \hat{P}_{t+1}(q) \ln \hat{P}_{t+1}(q) dq \\
&+ \mu \left( \int \hat{P}_{t+1}(q) dq - 1 \right),
\end{aligned}
\tag{9}
$$

in which we have included a Lagrange multiplier, μ, corresponding to the normalization constraint, $\int \hat{P}_{t+1}(q) dq = 1$. Taking the functional derivative of $L$ and setting to zero, we obtain

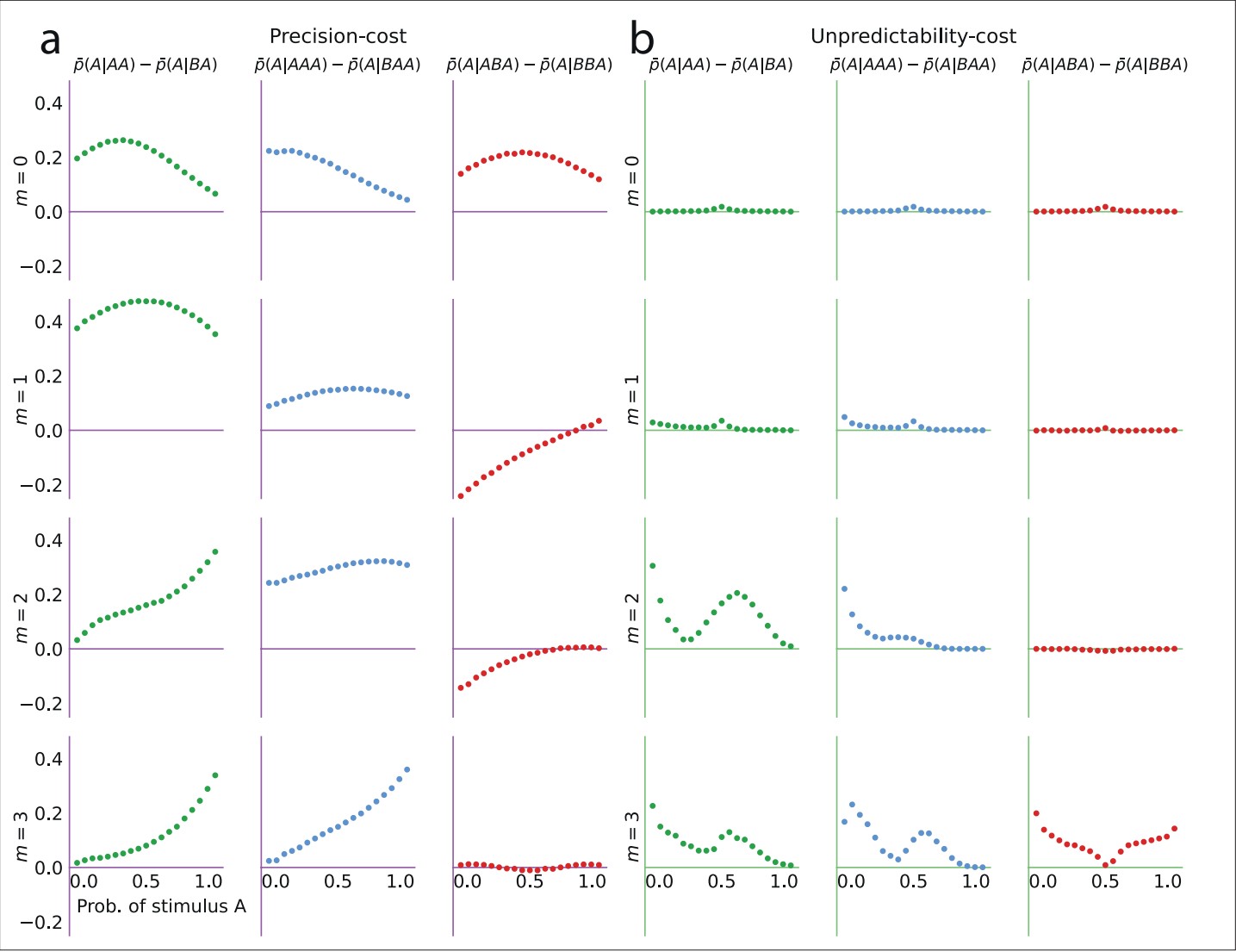

**Figure 8.** Sequential effects of the second- and third-to-last stimuli in precision-cost and unpredictability-cost models. (**a**) Precision-cost models. (**b**) Unpredictability-cost models. *First row*: Bernoulli observers (m = 0). *Second to fourth rows*: Markov observers (m = 1, 2, and 3). *First column (each panel)*: difference between the proportion of predictions A in the model subject's responses, conditional on the two preceding observations being the sequence AA, and the same proportion conditional on the sequence BA. A positive difference indicates an attractive sequential effect of the second-to-last stimulus. *Second column (each panel)*: difference between the proportion of predictions A in the model subject's responses, conditional on the three preceding observations being the sequence AAA, and the same proportion conditional on the sequence BAA. *Third column (each panel)*: difference between the proportion of predictions A in the model subject's responses, conditional on the three preceding observations being the sequence ABA, and the same proportion conditional on the sequence BBA. The precision-cost models of Markov observers are the only models that yield a negative difference, i.e., a repulsive sequential effect of the third-to-last stimulus, in this case.

$$\ln \hat{P}_{t+1}(q) + 1 - \ln P_{t+1}(q) + \lambda_u H(X; q) + \lambda_p \ln \hat{P}_{t+1}(q) + \lambda_p + \mu = 0, \tag{10}$$

and thus we write the approximate posterior as

$$\hat{P}_{t+1}(q) \propto P_{t+1}(q)^{\frac{1}{1+\lambda_p}} \exp\left(-\frac{\lambda_u}{1+\lambda_p} H(X; q)\right), \tag{11}$$

where $P_{t+1}(q)$ is the Bayesian update of the preceding belief, $\hat{P}_t(q)$, i.e.,

$$P_{t+1}(q) \propto \hat{P}_t(q) p(x_{t+1}|q, x_{t-m+1:t}). \tag{12}$$

Setting the weight of the unpredictability cost to zero (i.e., $\lambda_u = 0$), we obtain the posterior in presence of the precision cost only, as

$$\hat{P}_{t+1}^{prec}(q) \propto P_{t+1}(q)^{\frac{1}{1+\lambda_p}}. \tag{13}$$

The main text provides more details about the posterior in this case (*Equation 4*), in particular with a Bernoulli observer ($m = 0$; *Equation 5*, *Equation 6*).

For the hybrid model (in which both $\lambda_u$ and $\lambda_p$ are potentially different from zero), we obtain

$$\hat{P}_t(q) \propto \hat{P}_t^{prec}(q) \exp\left( -\lambda_u H(X; q) \sum_{i=1}^{t} \left( \frac{1}{1+\lambda_p} \right)^i \right). \tag{14}$$

With $\lambda_p = 0$, the sum in the exponential is equal to $t$, and the precision-cost posterior, $\hat{P}_t^{prec}(q)$, is the Bayesian posterior, $P_t^*(q)$, and thus we obtain the posterior in presence of the unpredictability cost only (see *Equation 8*).

## Hybrid models

The hybrid model, described above, features both a precision cost and an unpredictability cost, with respective weights $\lambda_p$ and $\lambda_u$. As with the models that include only one type of cost, we consider a Bernoulli observer ($m = 0$), and three Markov observers ($m = 1, 2,$ and $3$). As for the response-selection strategy, we use, here also, the generalized probability-matching strategy parameterized by $\kappa$. We thus obtain four new models; each one has three parameters ($\lambda_p$, $\lambda_u$, and $\kappa$), while the non-hybrid models (featuring only one type of cost) have only two parameters.

We fit these models to the responses of subjects. For 68% of subjects, the BIC of the best-fitting hybrid model is larger than the BIC of the best-fitting non-hybrid model, indicating a worse fit, by this measure. This suggests that for these subjects, allowing for a second type of cost result in a modest improvement of the fit that does not justify the additional parameter. For the remaining 32% of subjects, the hybrid models yield a better fit (a lower BIC) than the non-hybrid models, although for half of these, the difference in BICs is lower than 6, which is only weak evidence in favor of the hybrid models.

Moreover, we compute the exceedance probability, defined below in the section 'Bayesian Model Selection', of the hybrid models (together with the complementary probability of the non-hybrid models). We find that the exceedance probability of the hybrid models is 8.1% while that of the non-hybrid models is 91.9%, suggesting that subjects best-fitted by non-hybrid models are more prevalent.

In summary, we find that for more than two thirds of subjects, allowing for a second cost type does not improve much the fit to the behavioral data (the BIC is higher with the best-fitting hybrid model). These subjects are best-fitted by non-hybrid models, that is, by models featuring only one type of cost, instead of 'falling in between' the two cost types. This suggests that for most subjects, only one of the two costs, either the prediction cost or the unpredictability cost, dominates the inference process.

## Alternative response-selection strategy, and repetition or alternation propensity

In addition to the generalized probability-matching response-selection strategy presented in the main text, in our investigations we also implement several other response-selection strategies. First, a strategy based on a 'softmax' function that smoothes the optimal decision rule; it does not yield, however, a behavior substantially different from that of the generalized probability-matching response-selection strategy. Second, we examine a strategy in which the model subject chooses the optimal response with a probability that is fixed across conditions, which we fit onto subjects' choices. No subject is best-fitted by this strategy. Third, another possible strategy proposed in the game-theory literature (*Nowak and Sigmund, 1993*) is 'win-stay, lose-shift': it prescribes to repeat the same response as long as it proves correct and to change otherwise. In the context of our binary-choice prediction task, it is indistinguishable from a strategy in which the model subject chooses a prediction equal to the outcome that last occurred. This strategy is a special case of our Bernoulli observer hampered by a precision cost whose weight $\lambda$ is large combined with the optimal response-selection

strategy ($\kappa \to \infty$). Since the generalized probability-matching strategy parameterized by the exponent $\kappa$ appears either more general, better than or indistinguishable from those other response-selection strategies, we selected it to obtain the results presented in the main text.

Furthermore, we consider the possibility that subjects may have a tendency to repeat their preceding response, or, conversely, to alternate and choose the other response, independently from their inference of the stimulus statistics. Specifically, we examine a generalization of the response-selection strategy, in which a parameter $\eta$, with $-1 < \eta < 1$, modulates the probability of a repetition or of an alternation. With probability $1 - |\eta|$, the model subject chooses a response with the generalized probability-matching response-selection strategy, with parameter $\kappa$. With probability $|\eta|$, the model subject repeats the preceding response, if $\eta$ is positive; or chooses the opposite of the preceding response, if $\eta$ is negative. With $\eta = 0$, there is no propensity for repetition nor alternation, and the response-selection strategy is the same as the one we have considered in the main text. We have allowed for alternations ($\eta < 0$) in this model for the sake of generality, but for all the subjects the best-fitting value of $\eta$ is non-negative, thus henceforth we only consider the possibility of repetitions, i.e., non-negative values of the parameter ($\eta \geq 0$).

We note that with a repetition probability $\eta$, such that $0 \leq \eta < 1$, the unconditional probability of a prediction A, which we denote by $\bar{p}_\eta(A)$, is not different from the unconditional probability of a prediction A in the absence of a repetition probability $\eta$, $\bar{p}(A)$, as in the event of a repetition, the response that is repeated is itself A with probability $\bar{p}(A)$; formally, $\bar{p}_\eta(A) = (1 - \eta)\bar{p}(A) + \eta\bar{p}_\eta(A)$, which implies the equality $\bar{p}_\eta(A) = \bar{p}(A)$.

Now turning to sequential effects, we note that with a repetition probability $\eta$, the probability of a prediction $A$ conditional on an observation A is

$$\bar{p}_\eta(A|A) = (1 - \eta)\bar{p}(A|A) + \eta\bar{p}(A). \tag{15}$$

In other words, when introducing the repetition probability $\eta$, the resulting probability of a prediction A conditional on observing A is a weighted mean of the unconditional probability of a prediction A and of the conditional probability of a prediction A in the absence of a repetition probability. *Figure 7* (dotted lines) illustrates this for the eight models, with $\eta = 0.2$. Consequently the sequential effects with this response-selection strategy are more modest (*Figure 7*, light-red dots).

We fit (by maximizing their likelihoods) our eight models now equipped with a propensity for repetition (or alternation) parameterized by $\eta$. The average best-fitting value of $\eta$, across subjects, is 0.21 (standard deviation: 0.19; median: 0.18); as mentioned, no subjects have a negative best-fitting value of $\eta$. In order to assess the degree to which the models with repetition propensity are able to capture subjects' data, in comparison with the models without such propensity, we use the Bayesian Information Criterion (BIC) (*Schwarz, 1978*), which penalizes the number of parameters, as a comparative metric (a lower BIC is better). For 26% of subjects, the BIC with this response-selection strategy (allowing for $\eta \neq 0$) is higher than with the original response-selection strategy (which sets $\eta = 0$,) suggesting that the responses of these subjects do not warrant the introduction of a repetition (or alternation) propensity. In addition, for these subjects the best-fitting inference model, characterized by a cost type and a Markov order, is the same when the response-selection strategy allows for repetition or alternation ($\eta \neq 0$) and when it does not ($\eta = 0$). For 47% of subjects, the BIC is lower when including the parameter $\eta$ (suggesting that allowing for $\eta \neq 0$ results in a better fit to the data), and importantly, here also the best-fitting inference model (cost type and Markov order) is the same with $\eta \neq 0$ and with $\eta = 0$. For 11% of subjects, a better fit (lower BIC) is obtained with $\eta \neq 0$; and the best-fitting inference models, with $\eta \neq 0$ and with $\eta = 0$, belong to the same family of models, that is, they have the same cost type (precision cost or unpredictability cost), and only their Markov orders differ. Finally, only for the remaining 16% does the cost type change when allowing for $\eta \neq 0$. In other words, for 84% of subjects the best-fitting cost type is the same whether or not $\eta$ is allowed to differ from 0.

Furthermore, the best-fitting parameters $\lambda$ and $\kappa$ are also stable across these two cases. Among the 73% of subjects whose best-fitting inference model (including both cost type and Markov order) remains the same regardless of the presence of a repetition propensity, we find that the best-fitting values of $\kappa$, with $\eta \neq 0$ and with $\eta = 0$, differ by less than 10% for 93% of subjects, and the best-fitting values of $\lambda$ differ by less than 10% for 71% of subjects. For these two parameters, the correlation coefficient (between the best-fitting value with $\eta = 0$ and the best-fitting value with $\eta \neq 0$) is above 0.99 (with p-values lower than 1e-19).

The responses of a majority of subjects are thus better reproduced by a response-selection strategy that includes a probability of repeating the preceding response. The impact of this repetition propensity on sequential effects is relatively small in comparison to the magnitude of these effects (*Figure 7*). For most subjects, moreover, the best-fitting inference model, characterized by its cost type and its Markov order, is the same — with or without repetition propensity —, and the best-fitting parameters $\lambda$ and $\kappa$ are very close in the two cases. Therefore, this analysis supports the results of the model-fitting and model-selection procedure, and validates its robustness. We conclude that the models of costly inference are essential in reproducing the behavioral data, notwithstanding a positive repetition propensity in a fraction of subjects.

## Computation of the models' likelihoods

Model fitting is conducted by maximizing for each model the likelihood of the subject's choices. With the precision-cost models, the likelihood can be derived analytically and thus easily computed: the model's posterior is a Dirichlet distribution of order $2^{m+1}$, whose parameters are exponentially filtered counts of the observed sequences of length $m + 1$. With a Bernoulli observer, i.e., $m = 0$, this is the Beta distribution presented in *Equation 5*. The expected probability of a stimulus A, conditional on the sequence of $m$ stimuli most recently observed, is a simple ratio involving the exponentially filtered counts, for example $(\tilde{n}_t^A + 1)/(\tilde{n}_t^A + \tilde{n}_t^B + 2)$ in the case of a Bernoulli observer. This probability is then raised to the power $\kappa$ and normalized (as prescribed by the generalized probability-matching response-selection strategy) in order to obtain the probability of a prediction A.

As for the unpredictability-cost models, the posterior is given in *Equation 8* up to a normalization constant. Unfortunately, the expected probability of a stimulus A implied by this posterior does not come in a closed-form expression. Thus we compute the (unnormalized) posterior on a discretized grid of values of the vector $q$. The dimension of the vector $q$ is $2^m$, and each element of $q$ is in the segment $[0, 1]$. If we discretize each dimension into $n$ bins, we obtain $n^{2^m}$ different possible values of the vector $q$; for each of these, at each trial, we compute the unnormalized value of the posterior (as given by *Equation 8*). As $m$ increases, this becomes computationally prohibitive: for instance, with $n = 100$ bins and $m = 3$, the multidimensional grid of values of $q$ contains $10^{16}$ numbers (with a typical computer, this would represent 80,000 terabytes). In order to keep the needed computational resources within reasonable limits, we choose a lower resolution of the grid for larger values of $m$. Specifically, for $m = 0$ we choose a grid (over $[0, 1]$) with increments of 0.01; for $m = 1$, increments of 0.02 (in each dimension); for $m = 2$, increments of 0.05; and for $m = 3$, increments of 0.1. We then compute the mean of the discretized posterior and pass it through the generalized probability-matching response-selection model to obtain the choice probability.

To find the best-fitting parameters $\lambda$ and $\kappa$, the likelihood was maximized with the L-BFGS-B algorithm (*Byrd et al., 1995*; *Zhu et al., 1997*). These computations were run using Python and the libraries Numpy and Scipy (*Harris et al., 2020*; *Virtanen et al., 2020*).

## Symmetries and relations between conditional probabilities

Throughout the paper, we leverage the symmetry inherent to the Bernoulli prediction task to present results in a condensed manner. Specifically, in our analysis, the proportion of predictions A when the probability of A (the stimulus generative probability) is $p$, which we denote here by $\bar{p}(A|p)$, is equal to the proportion of predictions B when the probability of A is $1 - p$, which we denote by $\bar{p}(B|1 - p)$; i.e., $\bar{p}(A|p) = \bar{p}(B|1 - p)$. More generally, the predictions conditional on a given sequence when the probability of A is $p$ are equal to the predictions conditional on the 'mirror' sequence (in which A and B have been swapped), when the probability of A is $1 - p$, for example extending our notation, $\bar{p}(A|AAB, p) = \bar{p}(B|BBA, 1 - p)$. Here, we show how this results in the symmetries in *Figure 2*, and in the fact that in *Figures 5 and 6*, it suffices to plot the sequential effects obtained with only a fraction of all the possible sequences of two or three stimuli.

First, we note that

$$\bar{p}(A|p) = 1 - \bar{p}(B|p)$$
$$= 1 - \bar{p}(A|1 - p), \tag{16}$$

which implies the symmetry of $\bar{p}(A)$ in *Figure 2a* (grey line). Turning to conditional probabilities (and thus sequential effects), we have

$$\bar{p}(A|A,p) = 1 - \bar{p}(B|A,p)$$

$$= 1 - \bar{p}(A|B, 1-p) \tag{17}$$

$$\text{and } \bar{p}(A|B,p) = 1 - \bar{p}(A|A, 1-p).$$

As a result, the lines representing $\bar{p}(A|A)$ (blue) and $\bar{p}(A|B)$ (orange) in *Figure 2a* are reflections of each other. In addition, these equations result in the equality

$$\bar{p}(A|A,p) - \bar{p}(A|B,p) = \bar{p}(A|A, 1-p) - \bar{p}(A|B, 1-p), \tag{18}$$

which implies the symmetry in *Figure 2b*.

As for the sequential effect of the second-to-last stimulus, we show in *Figures 5a and 6a* the difference in the proportions of predictions A conditional on two past sequences of two stimuli, AA and BA; i.e., $\bar{p}(A|AA) - \bar{p}(A|BA)$. There are two other possible sequences of two stimuli: $AB$ and $BB$. The difference in the proportions conditional on these two sequences is implied by the former difference, as:

$$\bar{p}(A|AB,p) - \bar{p}(A|BB,p) = 1 - \bar{p}(B|AB,p) - (1 - \bar{p}(B|BB,p))$$

$$= 1 - \bar{p}(A|BA, 1-p) - (1 - \bar{p}(A|AA, 1-p)) \ . \tag{19}$$

$$= \bar{p}(A|AA, 1-p) - \bar{p}(A|BA, 1-p)$$

As for the sequential effect of the third-to-last stimulus, we show in *Figures 5b and 6b* the difference in the proportions conditional on the sequences AAA and BAA, and in *Figures 5c and 6c* the difference in the proportions conditional on the sequences ABA and BBA. The differences in the proportions conditional on the sequences AAB and BAB, and conditional on the sequences ABB and BBB, are recovered as a function of the former two, as

$$\bar{p}(A|AAB,p) - \bar{p}(A|BAB,p) = 1 - \bar{p}(B|AAB,p) - (1 - \bar{p}(B|BAB,p))$$

$$= 1 - \bar{p}(A|BBA, 1-p) - (1 - \bar{p}(A|ABA, 1-p))$$

$$= \bar{p}(A|ABA, 1-p) - \bar{p}(A|BBA, 1-p),$$

$$\text{and } \bar{p}(A|ABB,p) - \bar{p}(A|BBB,p) = 1 - \bar{p}(B|ABB,p) - (1 - \bar{p}(B|BBB,p)) \tag{20}$$

$$= 1 - \bar{p}(A|BAA, 1-p) - (1 - \bar{p}(A|AAA, 1-p))$$

$$= \bar{p}(A|AAA, 1-p) - \bar{p}(A|BAA, 1-p).$$

## Bayesian model selection

We implement the Bayesian model selection (BMS) procedure described in *Stephan et al., 2009*. Given $M$ models, this procedure aims at deriving a probabilistic belief on the distribution of these models among the general population. This unknown distribution is a categorical distribution, parameterized by the probabilities of the $M$ models, denoted by $r = (r_1, \ldots, r_M)$, with $\sum r_m = 1$. With a finite sample of data, one cannot determine with infinite precision the values of the probabilities $r_m$. The BMS, thus, computes an approximation of the Bayesian posterior over the vector $r$, as a Dirichlet distribution parameterized by the vector $\alpha = (\alpha_1, \ldots, \alpha_M)$, i.e., the posterior distribution

$$p(r|\alpha) = \frac{1}{Z(\alpha)} \prod_{m=1}^{M} r_m^{\alpha_m - 1}. \tag{21}$$

Computing the parameters $\alpha_k$ of this posterior makes use of the log-evidence of each model for each subject, i.e., the logarithm of the joint probability, $p(y|m)$, of a given subject's responses, $y$, under the assumption that a given model, $m$, generated the responses. We use the model's maximum likelihood to obtain an approximation of the model's log-evidence, as (*Balasubramanian, 1997*)

$$\ln p(y|m) \simeq \max_{\theta}[\ln p(y|m,\theta)] - \frac{d}{2} \ln N, \tag{22}$$

where $\theta$ denotes the parameters of the model, $p(y|m,\theta)$ is the likelihood of the model when parameterized with $\theta$, $d$ is the dimension of $\theta$, and $N$ is the size of the data, that is, the number of responses.

(The well-known Bayesian Information Criterion *Schwarz, 1978* is equal to this approximation of the model's log-evidence, multiplied by $-1/2$.)

In our case, there are $M = 8$ models, each with $d = 2$ parameters: $\theta = (\lambda, \kappa)$. The posterior distribution over the parameters of the categorical distribution of models in the general population, $p(r|\alpha)$, allows for the derivation of several quantities of interest; following *Stephan et al., 2009*, we derive two types of quantities. First, given a family of models, that is, a set $\mathcal{M} = \{m_i\}$ of different models (for instance, the prediction-cost models, or the Bernoulli-observer models), the expected probability of this class of model, that is, the expected probability that the behavior of a subject randomly chosen in the general population follows a model belonging to this class, is the ratio

$$\frac{\sum_{m \in \mathcal{M}} \alpha_m}{\sum_{m=1}^{K} \alpha_m}. \tag{23}$$

We compute the expected probability of the precision-cost models (and the complementary probability of the unpredictability-cost models), and the expected probability of the Bernoulli-observer models (and the complementary probability of the Markov-observer models; see Results).

Second, we estimate, for each family of models $\mathcal{M}$, the probability that it is the most likely, i.e., the probability of the inequality

$$\sum_{m \in \mathcal{M}} r_m > 1/2, \tag{24}$$

which is called the 'exceedance probability'. We compute an estimate of this probability by sampling one million times the Dirichlet belief distribution (*Equation 21*), and counting the number of samples in which the inequality is verified. We estimate in this way the exceedance probability of the precision-cost models (and the complementary probability of the unpredictability-cost models), and the exceedance probability of the Bernoulli-observer models (and the complementary probability of the Markov-observer models; see Results).

## Unpredictability cost for Markov observers

Here we derive the expression of the unpredictability cost for Markov observers as a function of the elements of the parameter vector $q$. For an observer of Markov order 1 ($m = 1$), the vector $q$ has two elements, which are the probability of observing A at a given trial conditional on the preceding outcome being A, and the probability of observing A at a given trial conditional on the preceding outcome being B, which we denote by $q_A$ and $q_B$, respectively. The Shannon entropy, $H(X; q)$, implied by the vector $q$, is the average of the conditional entropies implied by each conditional probability, i.e.,

$$H(X; q) = p_B H(X; q_B) + p_A H(X; q_A), \tag{25}$$

where $p_A$ and $p_B$ are the *unconditional* probabilities of observing A and B, respectively (see below), and

$$H(X; q_X) = -q_X \ln q_X - (1 - q_X) \ln(1 - q_X), \tag{26}$$

where $X$ is A or B.

The unconditional probabilities $p_A$ and $p_B$ are functions of the conditional probabilities $q_A$ and $q_B$. Indeed, at trial $t + 1$, the marginal probability of the event $x_{t+1} = A$, $P(x_{t+1} = A)$, is a weighted average of the probabilities of this event conditional on the preceding stimulus, $x_t$, as given by the law of total probability:

$$P(x_{t+1} = A) = P(x_{t+1} = A|x_t = A)P(x_t = A) + P(x_{t+1} = A|x_t = B)P(x_t = B), \tag{27}$$

i.e.

$$p_A = q_A p_A + q_B(1 - p_A). \tag{28}$$

Solving for $p_A$, we find:

$$p_A = \frac{q_B}{1 + q_B - q_A}, \text{ and } p_B = 1 - p_A = \frac{1 - q_A}{1 + q_B - q_A}. \tag{29}$$

The entropy $H(X; q)$ implied by the vector $q$ is obtained by substituting these quantities in *Equation 25*.

Similarly, for $m = 2$ and 3, the $2^m$ elements of the vector $q$ are the parameters $q_{ij}$ and $q_{ijk}$, respectively, where $i, j, k \in \{A, B\}$, and where $q_{ij}$ is the probability of observing A at a given trial conditional on the two preceding outcomes being the sequence '$ij$', and $q_{ijk}$ is the probability of observing A at a given trial conditional on the three preceding outcomes being the sequence '$ijk$'. The Shannon entropy, $H(X; q)$, implied by the vector $q$, is here also the average of the conditional entropies implied by each conditional probability, as

$$H(X; q) = \sum_{ij} p_{ij} H(X; q_{ij}), \text{ for } m = 2, \tag{30}$$

$$\text{and } H(X; q) = \sum_{ijk} p_{ijk} H(X; q_{ijk}), \text{ for } m = 3, \tag{31}$$

where $p_{ij}$ and $p_{ijk}$ are the unconditional probabilities of observing the sequence '$ij$', and of observing the sequence '$ijk$', respectively. These unconditional probabilities verify a system of linear equations whose coefficients are given by the conditional probabilities. For instance, for $m = 2$, we have the relation

$$
\begin{aligned}
P(x_t = A, x_{t+1} = A) &= P(x_{t-1} = A, x_t = A, x_{t+1} = A) + P(x_{t-1} = B, x_t = A, x_{t+1} = A) \\
&= P(x_{t-1} = A, x_t = A) P(x_{t+1} = A | x_{t-1} = A, x_t = A) \\
&\quad + P(x_{t-1} = B, x_t = A) P(x_{t+1} = A | x_{t-1} = B, x_t = A)
\end{aligned}
\tag{32}
$$

i.e.,

$$p_{AA} = p_{AA} q_{AA} + p_{BA} q_{BA}. \tag{33}$$

The system of linear equations can be written as

$$
\begin{pmatrix} p_{AA} \\ p_{AB} \\ p_{BA} \\ p_{BB} \end{pmatrix} =
\begin{pmatrix}
q_{AA} & 0 & q_{BA} & 0 \\
1 - q_{AA} & 0 & 1 - q_{BA} & 0 \\
0 & q_{AB} & 0 & q_{BB} \\
0 & 1 - q_{AB} & 0 & 1 - q_{BB}
\end{pmatrix}
\begin{pmatrix} p_{AA} \\ p_{AB} \\ p_{BA} \\ p_{BB} \end{pmatrix}.
\tag{34}
$$

The solution is the eigenvector corresponding to the eigenvalue equal to 1 of the matrix in the equation above, with the additional constraint that the unconditional probabilities must sum to 1, i.e., $\sum_{ij} p_{ij} = 1$. We find:

$$p_{BB} = \left( 1 - \frac{2 q_{BB}}{1 - q_{AB}} + \frac{q_{BA}}{1 - q_{AA}} \frac{q_{BB}}{1 - q_{AB}} \right)^{-1}$$

$$p_{BA} = \frac{q_{BB}}{1 - q_{AB}} p_{BB},$$

$$p_{AB} = p_{BA},$$

$$\text{and } p_{AA} = \frac{q_{BA}}{1 - q_{AA}} \frac{q_{BB}}{1 - q_{AB}} p_{BB}.$$

For $m = 3$, we find the relations:

$$p_{BBA} = p_{BBB} \frac{q_{BBB}}{1 - q_{ABB}},$$

$$p_{BAB} = p_{BBB} \frac{q_{BBB}}{1 - q_{ABB}} \frac{1 - q_{BBA}(1 - q_{AAB}) - q_{ABA} q_{AAB}}{1 - q_{BAB}(1 - q_{ABA}) - q_{ABA} q_{AAB}},$$

$$p_{BAA} = p_{BAB} \frac{q_{ABA} q_{BAB}}{1 - q_{ABA} q_{AAB}} + p_{BBA} \frac{q_{BBA}}{1 - q_{ABA} q_{AAB}},$$

$$p_{AAB} = p_{BAA},$$

$$p_{ABA} = p_{BAB} + p_{BAA} - p_{BBA},$$

$$p_{AAB} = p_{BAA},$$

$$\text{and } p_{AAA} = p_{BAA} \frac{q_{BAA}}{1 - q_{AAA}}.$$

Together with the normalization constraint $\Sigma_{ijk}p_{ijk} = 1$, these relations allow determining the eight unconditional probabilities $p_{ijk}$, and thus the expression of the Shannon entropy.

## Acknowledgements

We thank Doron Cohen and Michael Woodford for inspiring discussions. This work was supported by the Alfred P Sloan Foundation through grant G-2020–12680 and the CNRS through UMR8023. A.P.C. was supported by a Ph.D. fellowship of the Fondation Pierre-Gilles de Gennes pour la Recherche. We acknowledge computing resources from Columbia University's Shared Research Computing Facility project, which is supported by NIH Research Facility Improvement Grant 1G20RR030893-01, and associated funds from the New York State Empire State Development, Division of Science Technology and Innovation (NYSTAR) Contract C090171, both awarded April 15, 2010.

## Additional information

### Funding

| Funder | Grant reference number | Author |
|---|---|---|
| Albert P. Sloan Foundation | Grant G-2020-12680 | Rava Azeredo da Silveira |
| CNRS | UMR8023 | Rava Azeredo da Silveira |
| Fondation Pierre-Gilles de Gennes pour la recherche | Ph.D. Fellowship | Arthur Prat-Carrabin |

The funders had no role in study design, data collection and interpretation, or the decision to submit the work for publication.

### Author contributions

Arthur Prat-Carrabin, Conceptualization, Data curation, Software, Formal analysis, Investigation, Visualization, Methodology, Writing – original draft, Project administration, Writing – review and editing; Florent Meyniel, Conceptualization, Formal analysis, Visualization, Methodology, Writing – review and editing; Rava Azeredo da Silveira, Conceptualization, Formal analysis, Supervision, Funding acquisition, Visualization, Methodology, Project administration, Writing – review and editing

### Author ORCIDs

Arthur Prat-Carrabin https://orcid.org/0000-0001-6710-1488
Florent Meyniel https://orcid.org/0000-0002-6992-678X
Rava Azeredo da Silveira https://orcid.org/0000-0002-8487-4105

### Ethics

The study was approved by the ethics committee Île de France VII (CPP 08-021). Participants gave their written consent prior to participating.

### Decision letter and Author response

Decision letter https://doi.org/10.7554/eLife.81256.sa1
Author response https://doi.org/10.7554/eLife.81256.sa2

## Additional files

### Supplementary files
• MDAR checklist

## Data availability

The behavioral data for this study and the computer code used for data analysis are freely and publicly available through the Open Science Framework repository at https://doi.org/10.17605/OSF.IO/BS5CY.

The following dataset was generated:

| Author(s) | Year | Dataset title | Dataset URL | Database and Identifier |
|---|---|---|---|---|
| Prat-Carrabin A, Meyniel F, Azeredo da Silveira R | 2022 | Resource-Rational Account of Sequential Effects in Human Prediction: Data & Code | https://doi.org/10.17605/OSF.IO/BS5CY | Open Science Framework, 10.17605/OSF.IO/BS5CY |

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

## Appendix 1

### Stability of subjects' behavior throughout the experiment

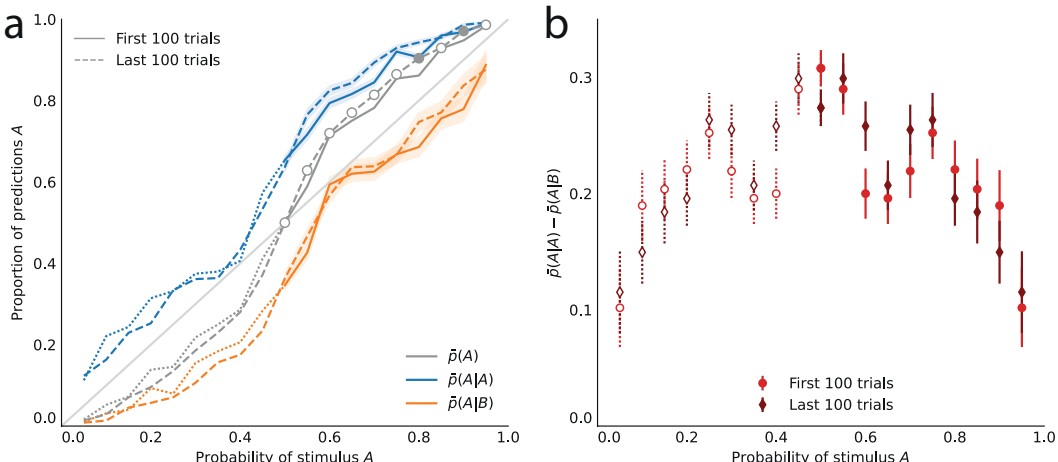

**Appendix 1—figure 1.** Subjects' behavior in the first and second halves of the task. (**a**) Proportion of predictions A as a function of the stimulus generative probability, conditional on observing A (blue lines) or B (orange lines), and unconditional (grey lines), in the first half of the experiment (solid lines) and in the second half (dashed lines). Filled circles indicate p-values of Fisher's exact test (of the equality of the proportions in the first and second halves, with Bonferroni-Holm-Šidák correction) below .05. (**b**) Difference between the proportions of predictions A conditional on an A, and conditional on a B, in the first half of the experiment (red circles), and in the second half (dark-red diamonds). The p-values of Fisher's exact tests (of equality of the conditional proportions, i.e., $\bar{p}(A|A) = \bar{p}(A|B)$), with Bonferroni-Holm-Šidák correction, are all below 1e-6. Bars indicate the standard error of the mean.

To validate the assumption that we capture, in our experiment, the 'stationary' behavior of subjects, we compare their responses in the first half of the task (first 100 trials) to their responses in the second half (last 100 trials). We find that the unconditional proportions of prediction A in these two cases are not significantly different, for most values of the stimulus generative probability. The sign of the difference (regardless of its statistical significance) implies that the proportions of predictions A in the second half of the experiment are slightly closer to 1 when the probability of the stimulus A is greater than 0.5; which would mean that the responses of subjects are slightly closer to optimality, in the second half of the experiment (*Appendix 1—figure 1a*, grey lines). Regarding the sequential effects, we also obtain very similar behaviors in the first and second halves of the experiment (*Appendix 1—figure 1*). We conclude that for our analysis it is reasonable to assume that the behavior of subjects is stationary throughout the task.

### Robustness of the model fitting

To evaluate the ability of the model-fitting procedure to correctly identify the model that generated a given set of responses, we compute a confusion matrix of the eight models. For each model, we simulate 200 runs of the task (each with 200 passive trials followed by 200 trials in which a prediction is obtained), with values of $\lambda$ and $\kappa$ close to values typically obtained when fitting the subjects' responses (for prediction-cost models, $\lambda \in \{0.03, 0.7, 2, 15\}$; for unpredictability-cost models, $\lambda \in \{0.7, 2\}$; and $\kappa \in \{0.7, 1.5, 2\}$ for both families of models). We then fit each of the eight models to each of these simulated datasets, and count how many times each model best fit each dataset (*Appendix 1—figure 2a*). To further test the robustness of the model-fitting procedure, we randomly introduce errors in the simulated responses: for 10% of the responses, randomly chosen in each dataset, we substitute the response by its opposite (i.e., B for A, and A for B), and compute a confusion matrix using these new responses (*Appendix 1—figure 2b*). In both cases, the model-fitting procedure identifies the correct model a majority of times (i.e., the best-fitting model is the model that generated the data; *Appendix 1—figure 2*).

Finally, to examine the robustness of the weight of the cost, $\lambda$, we consider for each subject its best-fitting model in each family (the precision-cost family and the unpredictability-cost family), and we fit separately each model to the subject's responses obtained in trials in which the stimulus

generative probability was medium ($p \in \{.3, .35, .4, .45, .5, .55, .6, .65, .7\}$) and those in which it was extreme ($p \in \{.05, .1, .15, .2, .25, .75, .8, .85, .9, .95\}$). The *Appendix 1—figure 3* shows the correlation between the best-fitting parameters obtained in these two cases.

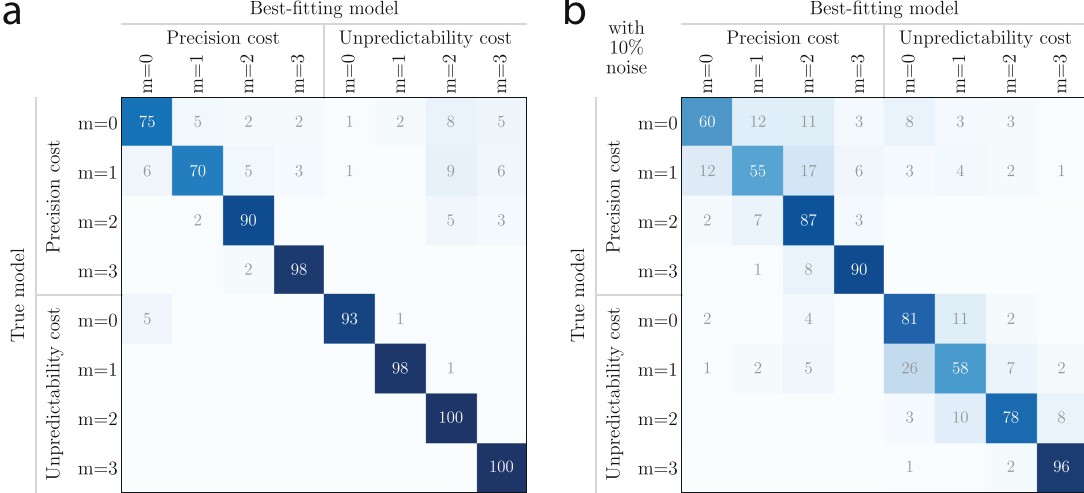

**Appendix 1—figure 2.** Model-fitting confusion matrix. (**a**) For each row models ('true model'), percentage of simulated datasets of 200 responses that were best fitted by column models ('best-fitting model'). Example: when fitting data generated by the precision-cost model with $m = 3$, the best-fitting model was the correct model on 98% of the fits, and the precision-cost model with $m = 2$ on 2% of the fits. (**b**) Same as (**a**), with 10% of responses (randomly chosen in each simulated dataset) replaced by the opposite responses.

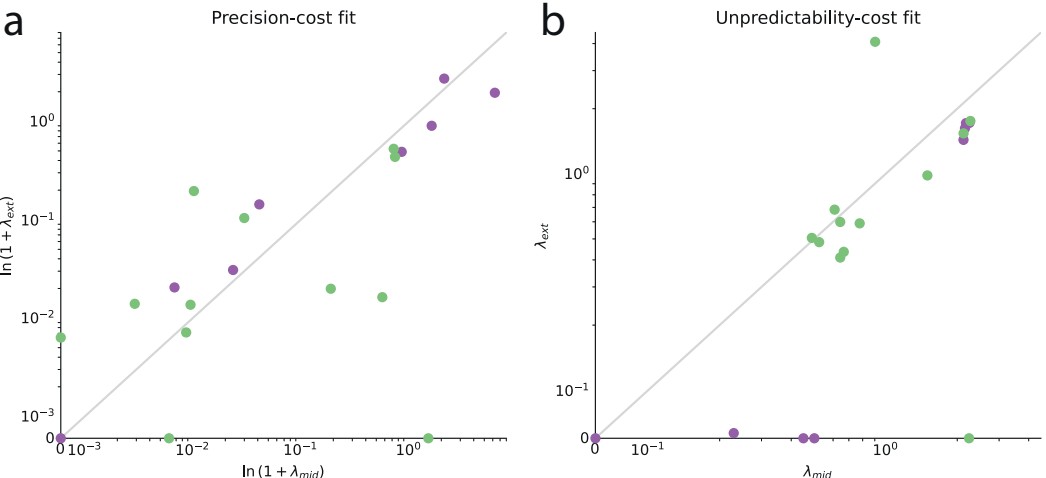

**Appendix 1—figure 3.** Stability of the cost-weight parameter across medium and extreme values of the stimulus generative probability. Best-fitting parameters of individual subjects when fitting the data obtained in trials with extreme values of the stimulus generative probability (i.e., $p$ or $1 - p$ in $\{.75, .8, .85, .9, .95\}$), plotted against the best-fitting parameters when fitting the data obtained in trials with medium values of the stimulus generative probability (i.e., $p$ or $1 - p$ in $\{.5, .55, .6, .65, .7\}$), with (**a**) precision-cost models, and (**b**) unpredictability-cost models. *Purple dots:* subjects best-fitted by prediction-cost models. *Green dots:* subjects best-fitted by unpredictability-cost models. The plots are in log-log scale, except below $10^{-3}$ (**a**) and $10^{-1}$ (**b**), where the scale is linear (allowing in particular for the value 0 to be plotted.) For the precision-cost models, we plot the inverse of the characteristic decay time, $\ln(1 + \lambda)$. The grey line shows the identity function.

## Distribution of subjects' BICs

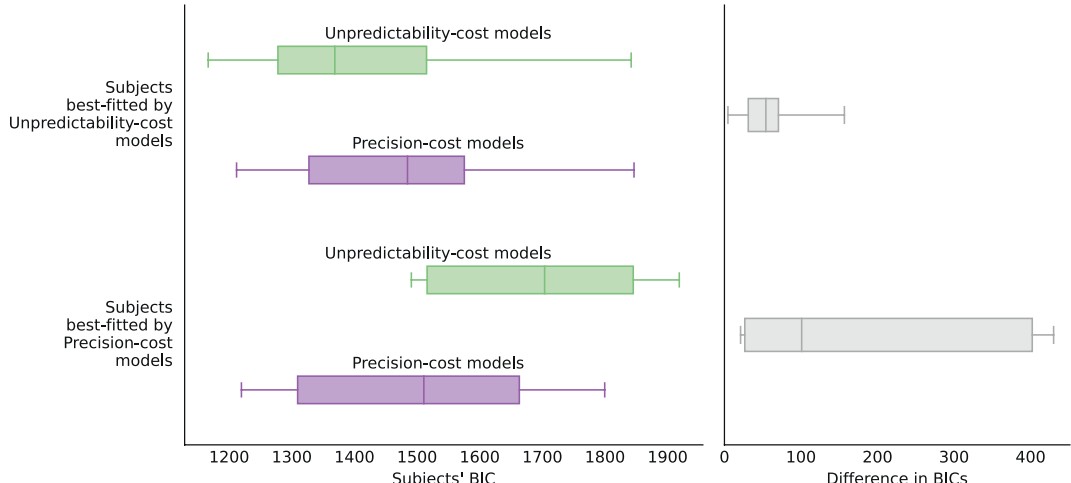

**Appendix 1—figure 4.** Distribution of subjects' BICs. (*Left*) Box-and-whisker plots showing the 5th and 95th percentiles (whiskers), the first and third quartiles (box), and the median (vertical line) of the BICs (across subjects) of the unpredictability-cost models (green boxes) and of the precision-cost models (purple boxes), fitted on the subjects best-fitted by the unpredictability-cost models (first two rows) and on the subjects best-fitted by the precision-cost models (last two rows). (*Right*) Box-and-whisker plots (same quantiles) showing the distribution of the difference, for each subject, between the BIC of the best model in the family that does not best fit the subject, and the BIC of the best-fitting model; for the subjects best-fitted by the unpredictability-cost models (top box) and for the subjects best-fitted by the precision-cost models (bottom box). The unpredictability-cost models, when fitted to the responses of the subjects best-fitted by the precision-cost models (bottom box), yield larger differences in the BIC with the best-fitting models, than the precision-cost models when fitted to the responses of the subjects best-fitted by the unpredictability-cost models (top box). This suggests that the precision-cost models are better than the unpredictability-cost models at capturing the responses of the subjects that they do not best fit.

## Subjects' sequential effects — tree representation

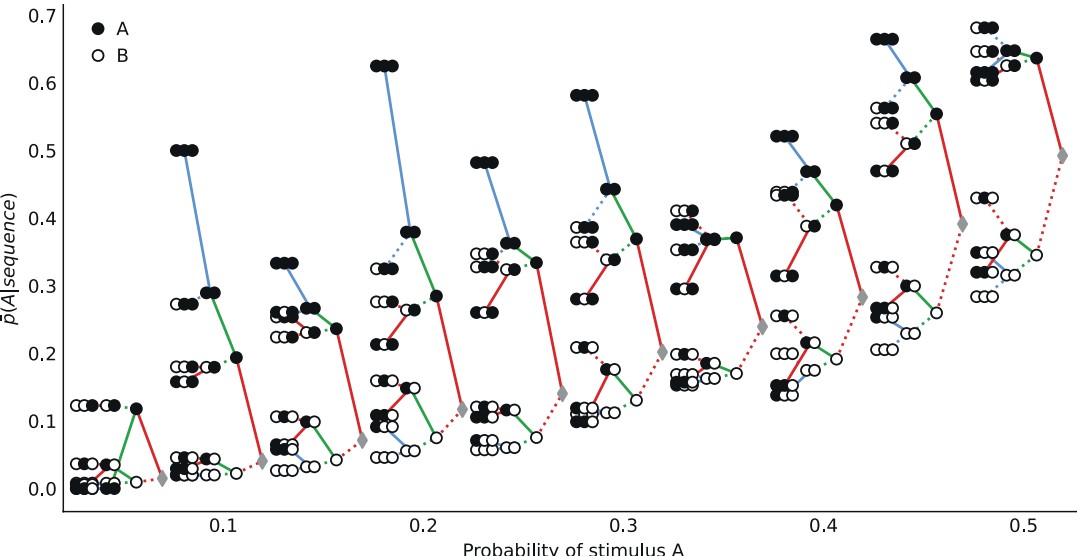

**Appendix 1—figure 5.** Composition of sequential effects in subjects' responses. Proportion of predictions A conditional on sequences of up to three past observations, as a function of the stimulus generative probability. See *Figure 6d*.

## Subjects' sequential effects — unpooled data

As mentioned in the main text, we pool together the predictions that correspond, in different blocks of trials, to either event (left or right), as long as these events have the same probability. The *Appendix 1—figure 6*, below, is the same as *Figure 2*, but without such pooling. Given a stimulus generative probability, $p$, all the subjects experience one (and only one) block of trials in which either the event 'right' or the event 'left' had probability $p$. For one group of subjects the 'right' event has probability $p$ and for the group of remaining subjects it is the 'left' event that has probability $p$. The responses of these subjects are not pooled together in *Appendix 1—figure 6*, while they were in *Figure 2*. This also applies for any other stimulus generative probability, $p'$. However, we note that the two groups of subjects for whom $p'$ was the probability of a 'right' event or a 'left' event are not the same as the two groups just mentioned in the case of the probability $p$. As a result, from one proportion shown in *Appendix 1—figure 6* to another, the underlying group of subjects changes. In *Figure 2*, each proportion is computed with the responses of all the subjects. This illustrates another advantage of the pooling that we use in the main text.

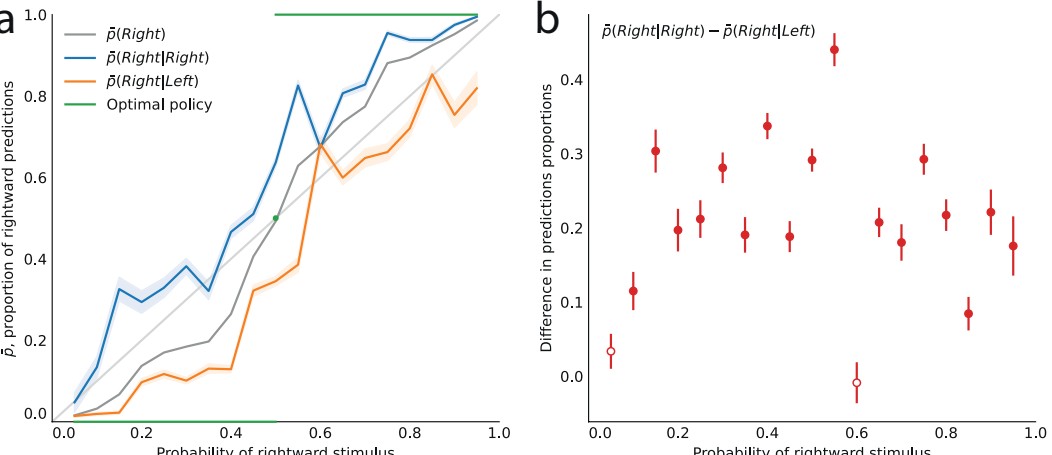

**Appendix 1—figure 6.** Sequential effects in subjects' responses. As *Figure 2*, but without pooling together the rightward and leftward predictions from different block of trials in which the corresponding stimuli have the same probability. See main text.

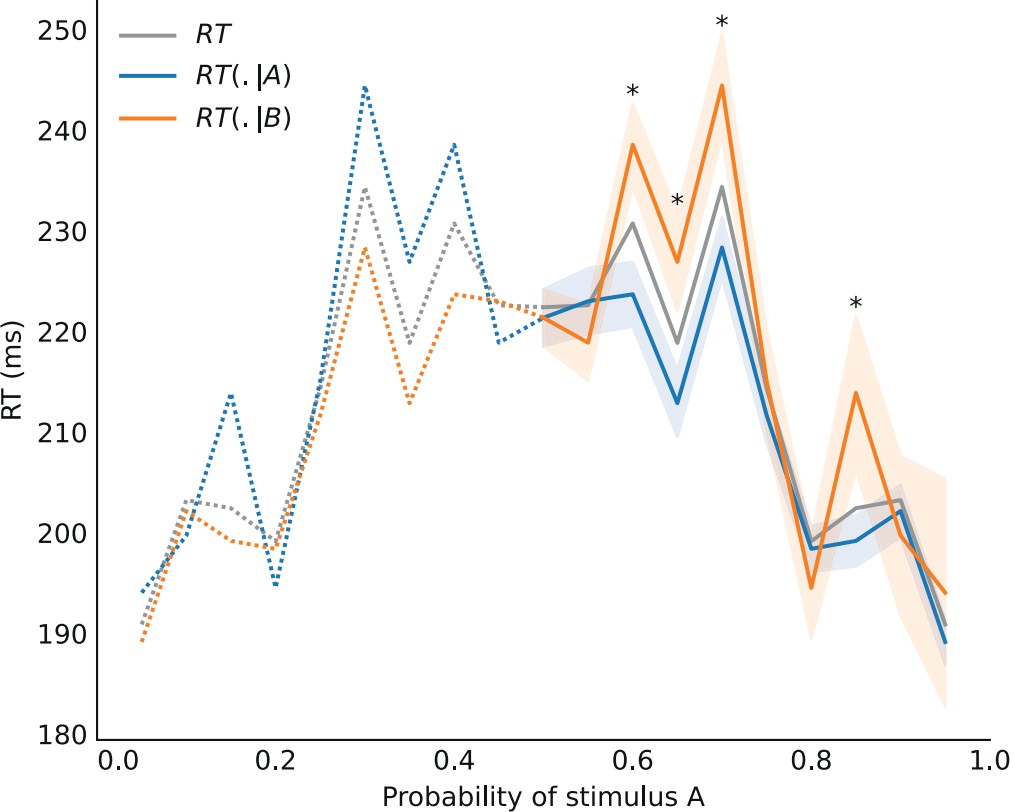

**Appendix 1—figure 7.** Subjects' response times. Average response times conditional on having observed a stimulus A (blue line), a stimulus B (orange line), and unconditional (grey line), as a function of the stimulus generative probability. The stars indicate that the p-values below 0.05 of the $t$-tests of equality between the response times after an A and after a B. The subjects seem slower after observing the less frequent stimulus (e.g., B, when $p > .5$).

## Across-subjects results

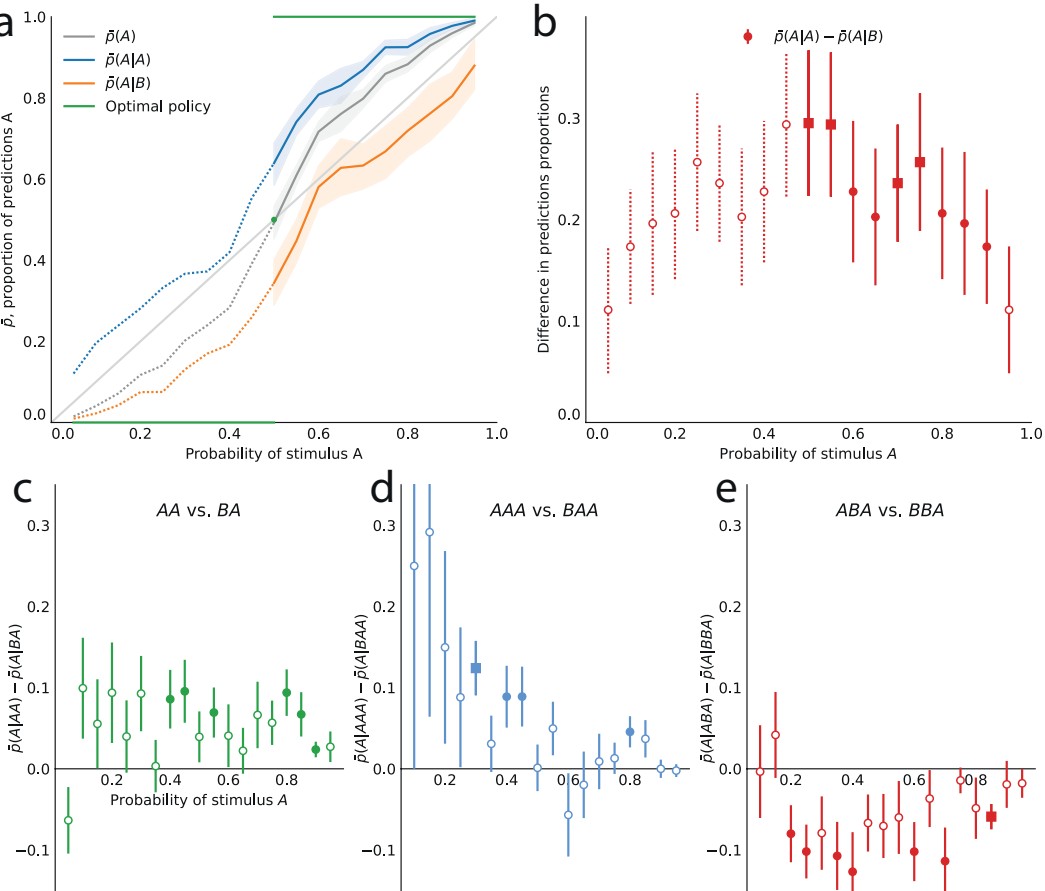

**Appendix 1—figure 8.** Subjects' sequential effects — across-subjects analysis. The statistics used for this figure are obtained by computing first the proportions of predictions A for each subject, and then computing the across-subject averages and standard errors of the mean (instead of pooling together the responses of all the subjects). (**a,b**) As in **Figure 2**, with across-subjects statistics. (**c,d,e**) As in **Figure 6a, b and c**, with across-subjects statistics. The filled circles indicate that the p-value of the Student's t-test is below 0.05, and the filled squares indicate that the p-value with Bonferroni-Holm-Šidák correction is below 0.05.

