## [Editor Report]

This valuable work addresses a long-standing empirical puzzle from a new computational perspective. The authors provide convincing evidence that attractive and repulsive sequential effects in perceptual decisions may emerge from rational choices under cognitive resource constraints rather than adjustments to changing environments. It is relevant to understanding how people represent uncertain events in the world around them and make decisions, with broad applications to economic behavior.

---

## [Decision Letter]

**Decision letter after peer review:**

Thank you for submitting your article "Resource-Rational Account of Sequential Effects in Human Prediction" for consideration by *eLife*. Your article has been reviewed by 3 peer reviewers, and the evaluation has been overseen by a Reviewing Editor and Floris de Lange as the Senior Editor. The reviewers have opted to remain anonymous.

Essential revisions:

1) Including alternative models of sequential effects in the model comparison would be necessary. Please see Reviewers #2's and #3's comments for details.

2) Additional statistical tests are required to tease out potentially confounding effects of motor responses (see Reviewer #3's comments). Besides, there should be corrections for multiple comparisons (see Reviewer #2's comments).

3) The costs assumed in the resource-rational models need better theoretical justification. Please see Reviewers #1's and #3's comments for details.

4) Testing a hybrid model that combines the precision cost and unpredictability cost is highly recommended, given that the two models seem to explain complementary aspects of the data. Please see Reviewer #1's comments for details.

*Reviewer #1 (Recommendations for the authors):*

There is a clear inverted-U shape in Figure 2b that the authors don't comment on. This seems like a salient feature of the data that should be explained or at least commented on. Interestingly, the best-fitting models can account for this (Figure 4b), but it doesn't seem to be discussed. It would also be helpful to see the predictions separately for precision-cost and unpredictability-cost models.

At a conceptual level, I'm not sure I understand the reasoning behind the unpredictability cost. It's not intuitive to me why more unpredictability should be registered by an observer as a cost of updating their beliefs. The Discussion didn't really clear this up; there's a reference to a preference for predictable environments, but the argument that this is somehow costly to the brain is hand-wavy. Just because unpredictability increases neural activity doesn't mean that it's something the brain is trying to minimize.

The pattern of higher-order sequential effects (Figure 6) seems to suggest that behavior is consistent with some combination of precision cost and unpredictability cost. Neither model on its own explains the data particularly well (compare with Figure 5). Have the authors considered hybrid models?

There are a few references related to the costly inference that deserve mention:

– Kominers et al. (2016), who develop a model of costly updating.

– Dasgupta et al. (2020), who develop a model of resource-bounded inference.

– Graeber (202), who develops a model of inattentive inference.

The authors cite a number of earlier modeling papers, but it's not clear to me what those previous models would predict about the new data, and whether the new models proposed in this paper predict the earlier data.

*Reviewer #2 (Recommendations for the authors):*

I would recommend the authors conduct additional modeling analyses including models that express the alternative hypotheses clearly, generate and present individual model predictions, and consider running a (possibly online) larger experiment with incentives in place. The ideal test of the hypothesis would be an experiment with multiple levels of incentives, showing that people adjust their representations as this model predicts as they trade off computational costs and real rewards. If monetary rewards cannot be used, incentives could be implemented by having a longer delay after erroneous predictions.

*Reviewer #3 (Recommendations for the authors):*

1. I recommend using regression analyses (e.g. a GLM) where regressors for both previous choices and previous stimuli are entered (as e.g. in Urai et al. Nature Communications 2017) to resolve this possible confound. Such GLM also allows looking back further back in time at the impact of longer lags on decisions. This would allow testing for example if the repulsive bias to the stimulus at lag 2 (in sequences 111 or 101) extends to longer lags, both in experimental data and simulations.

2. Authors could test whether the very first prediction on each block already shows a signature of the p and whether the prediction is stable within blocks.

I provide here some recommendations on how the clarity of the manuscript could be improved. A thorough work on improving the text and figures and the general flow and organization of the manuscript would make a major difference in the impact of that paper on the wider community. which is very frustrating for the reader is to see one statement (e.g. a choice of methods or a result) exposed at some point and then the explanation for the statement much later in the manuscript (see below). Here are my suggestions:

– I believe the models would be better motivated to the reader if the Introduction made a brief mention of the ideas of bounded rationality (and related concepts) and justified focus on these two specific types of cost – all of which are nicely detailed in the Discussion.

– Please try to make understanding figures more intuitive; for example, using a colour code for the different cost types may help differentiate them. A tree-like representation of history biases (showing the mean accuracy for different types of sequences, e.g. in Meyniel et al. 2016 Plos CB) may be more intuitive to read and reveal a richer structure in the data and models than the current Figure 5-6 (also given than the authors do not comment much on the impact of the "probability of observation 1", so perhaps this effects could be marginalized out).

– Figure 3 is really helpful in understanding the two types of cost (much more than the equations for most readers). Unfortunately, it is hardly referred to in the main text. I suggest rewriting the presentation of that part of the Results section around these examples.

– Why and how the two types of costs give rise to historical effects (beyond the fact that these costs generate suboptimalities) is a central idea in the paper but it is only exposed in the Discussion session. Integrating these explanations within the Results section would help a lot. Plotting some example simulations for a sequence of trials and/or some cartoon explanations of how the historical effects emerge for the different models would also help.

– Placing figures in Methods does not help in my opinion – please consider moving to the main text or as supplementary Figures.

---

## [Author Response]

Essential revisions:Reviewer #1 (Recommendations for the authors):There is a clear inverted-U shape in Figure 2b that the authors don't comment on. This seems like a salient feature of the data that should be explained or at least commented on. Interestingly, the best-fitting models can account for this (Figure 4b), but it doesn't seem to be discussed. It would also be helpful to see the predictions separately for precision-cost and unpredictability-cost models.

We agree with Reviewer #1 that it is notable that there is an inverted U shape in Figure 2b, i.e., that the sequential effect of the last stimulus is smaller for more extreme values of the stimulus generative probability, in the responses of the subjects. Some models reproduce this pattern: following Reviewer #1’s suggestion, we now show in Figure 4 the predictions of the precision-cost and unpredictability cost models separately.

The panels a, b, c, and d in Figure 4 show the predictions of the precision-cost model of a Bernoulli observer (a) and of a Markov observer with m=1 (c), and the unpredictability-cost model of a Bernoulli observer (b) and of a Markov observer with m=1 (d). In panel (a) we also show the predictions of the precision-cost model of a Bernoulli observer (m=0) with the “traditional” probability-matching strategy, i.e., with kappa = 1 (in this case the probability of predicting A is equal to the inferred probability of the event A). In this case the size of the sequential effect, p(A|A)-p(A|B), is the same for all values of stimulus generative probability (see dotted lines and light-red dots). But with kappa = 2.8 (a value representative of subjects’ best-fitting values), the proportions of predictions are brought closer to optimality (i.e., to the extremes), and the sequential effect, p(A|A)-p(A|B) now depends on the stimulus generative probability, resulting in the inverted U-shape of the sequential effects in Figure 4a.

We note, however, that the precision-cost model of a Markov observer (with m=1) yields a (non-inverted) U shape of the sequential effects. While the behavior of the precision-cost model of a Bernoulli observer is determined by two exponentiallyfiltered counts of the two possible stimuli, the behavior of the precision-cost model of a Markov observer (m=1) is determined by four exponentially-filtered counts of the four possible *pairs* of stimuli, and in particular p(A|B) is determined by the counts of the pairs BA and BB. But when p is large, the pairs BA and BB are rare: thus it is as if the model subject had little total evidence to inform its decision. The resulting predictions are close to that of an uninformed observer, i.e., p(A|B) ≈ 0.5. By contrast, p(A|A) is more extreme, and this difference yields stronger sequential effects for more extreme values of the stimulus generative probability (i.e., U shape in Figure 4c, right).

Among the subjects that are best-fitted by a precision-cost model, some are bestfitted by a model of a Bernoulli observer (m=0) while some others are best-fitted by a model of a Markov observer (m>0). Overall the sequential effects for these subjects exhibit a small decrease at more extreme stimulus generative probabilities (Figure 4e, right). By contrast, the subjects best-fitted by an unpredictability-cost model show a stronger decrease in the sequential effects at more extreme probabilities (Figure 4f, right). In addition the latter subjects exhibit weaker sequential effects than the former ones. This is reproduced by simulations of the corresponding best-fitting models (Figure 4g,h). The models belonging to the ‘other’ family, which does not best fit each subject (i.e., the precision-cost models, for the subjects best-fitted by an unpredictability-cost model; and the unpredictabilitycost models, for the subjects that are best-fitted by a precision-cost model) do not reproduce well the patterns in subjects’ data (Figure 4i,j).

In the revised version of the manuscript, we now point to the inverted U-shape of sequential effects in group average of subjects’ data (l. 182-185), and we have completely reworked the presentation of the sequential effects of the model. In particular we detail the behavior of each family of models, separately, using the updated Figure 4; we explain the origin of the shape of the sequential effects (inverted U-shape in Figure 4a and U-shape in Figure 4c); and we compare the models’ behaviors with that of the subjects (l. 422-493).

At a conceptual level, I'm not sure I understand the reasoning behind the unpredictability cost. It's not intuitive to me why more unpredictability should be registered by an observer as a cost of updating their beliefs. The Discussion didn't really clear this up; there's a reference to a preference for predictable environments, but the argument that this is somehow costly to the brain is hand-wavy. Just because unpredictability increases neural activity doesn't mean that it's something the brain is trying to minimize.

In the revised version of the manuscript, we now provide more details on the rationale for the unpredictability cost. We note, first, that we make a similar argument based on the cost of neural activity for the precision cost. Although the two cases differ by the hypothesized origin of the increase in the neural activity, in both cases we assume that this neural activity comes with a metabolic cost, which is not to be minimized per se, but which enters a trade-off with the correctness of the represented belief, in comparison to the optimal, Bayesian belief.

As to the rationale subtending the unpredictability cost, it resides in the assumption that the difficulty of representing a belief distribution over the parameters generating the environment (here, q), originates in the difficulty of representing the environments themselves. For instance, in the models of ‘intuitive physics’ (e.g., Battaglia, Hamrick, and Tenenbaum, 2013) or in the ‘simulation heuristic’ of Kahneman and Tversky (1982), the brain runs simulations of the possible sequences of outcomes, in a given environment. Environments that are more entropic result in a greater diversity of sequences of outcomes, and thus in more simulations, resulting, presumably, in higher costs. Furthermore, several cognitive models posit that the brain *compresses* sequences (Simon, 1972; Planton *et al.*, 2021); but a greater entropy in sequences reduces the compression rate, resulting in longer descriptions of these sequences (here also, because of the greater diversity of potential outcomes), which presumably is more costly.

We note that for neither the precision cost nor the unpredictability cost do we provide a mechanistic account of the underlying representational system in which the cost naturally emerges. But under the assumption that the cost of representing a distribution over environments resides in the cost of representing the environments themselves, it seems that, for the reasons just presented, a reasonable assumption is that more unpredictable environments are more difficult to represent.

In the revised version of the manuscript, we now provide more details on the rationale for the unpredictability cost, in the Discussion (l. 687-700). We have also reworked the short presentation of the costs in the Introduction (l. 71-79).

References:

Peter W. Battaglia, Jessica B. Hamrick, and Joshua B. Tenenbaum. Simulation as an engine of physical scene understanding. Proceedings of the National Academy of Sciences of the United States of America, 110(45):18327–18332, 2013.

Daniel Kahneman and Amos Tversky. The simulation heuristic, pages 201–208. Cambridge University Press, 1982.

Herbert A Simon. Complexity and the representation of patterned sequences of symbols. Psychological review, 79(5):369, 1972.

Samuel Planton, Timo van Kerkoerle, Leïla Abbih, Maxime Maheu, Florent Meyniel, Mariano Sigman, Liping Wang, Santiago Figueira, Sergio Romano, and Stanislas Dehaene. A theory of memory for binary sequences: Evidence for a mental compression algorithm in humans, 2021.

The pattern of higher-order sequential effects (Figure 6) seems to suggest that behavior is consistent with some combination of precision cost and unpredictability cost. Neither model on its own explains the data particularly well (compare with Figure 5). Have the authors considered hybrid models?

Regarding the patterns of sequential effects in subjects’ data and resulting from the models, we note that the main objective of Figure 5 was to illustrate the signs of the sequential effects occurring with the models, and in particular that the sequential effects are repulsive only with the precision-cost model of a Markov observer (m=1). A diversity of behaviors, however, can result from the models, depending on the type of cost (precision or unpredictability), on the Markov order (m=0 to 3), and on the values of the model’s parameters (λ and kappa). The Figure 8, in Methods, shows the higher-order sequential effects for the two types of costs and the four Markov orders we consider: depending on the model, the sequential effects can be an increasing function of the stimulus generative probability, or a decreasing function, or a non-monotonous function; but in all these cases the signs of the sequential effects are consistent with what is shown in Figure 5 and with the message we seek to convey, that in most cases the sequential effects are attractive, except in one case with the precision-cost model of a Markov observer (m>0).

Taking into account Reviewer #1’s comment, however, and for the benefit of the reader, we have added the behavior of another model to Figure 5, in the revised version of the manuscript: that of the precision-cost model of a Bernoulli observer (m=0). Not only does it exhibit how this model does not yield repulsive sequential effects (unlike the precision-cost model of a Markov observer, m=1; Figure 5c), but also it shows how it yields attractive sequential effects, in Figure 5a and 5b, whose behaviors as a function of the stimulus generative probability are qualitatively different from that of the precision-cost model of a Markov observer (m=1), thus suggesting the diversity of behaviors resulting from the models.

However, we agree with Reviewer #1 that hybrid models are an interesting possibility. Thus, we have investigated a hybrid model, in which both the precision cost and the unpredictability cost weigh on the representation of posteriors, each with a different weight parameter (denoted by λ_p_ and λ_u_). We derive the optimal inference procedure under this double cost, and find that it results in a posterior that fluctuates with the recent history of stimuli, and that is biased toward values of the generative parameter q that implies less entropic environments; in other words, it combines features of the two costs taken separately. For a given Markov order, m, this model is a generalization of both the precision-cost model and the unpredictability-cost model. It has one more parameter than these models (due to the two weights of the costs). To compare the ability of models to capture parsimoniously subjects’ data, we use as a comparison metric the well-known Bayesian Information Criterion (BIC), which is based on the log-likelihood but also includes a penalty term for the number of parameters. Although one might expect that the behavior of most subjects may be best captured (as per the BIC) by this hybrid model, we find that its BIC is in fact larger (indicating a worse fit) than the best-fitting non-hybrid model for more than two thirds of subjects; and for half of the remaining subjects (for whom the BIC is lower with the hybrid model), the difference in BIC is lower than 6, which indicates weak evidence in support of the hybrid models. In other words, for a majority of subjects, the improvement in the log-likelihood that results from allowing a second type of cost is too modest to justify the additional parameter. This suggests that the two families of models capture specific behaviors that are prevalent in different subpopulations of subjects. In the revised manuscript, we comment on the hybrid model in the main text (l. 407-419) and we present it in more detail in Methods (p. 44-46).

There are a few references related to the costly inference that deserve mention:– Kominers et al. (2016), who develop a model of costly updating.– Dasgupta et al. (2020), who develop a model of resource-bounded inference.– Graeber (202), who develops a model of inattentive inference.

We thank Reviewer #1 for pointing to these papers which also consider the hypothesis of a cost in the inference process of decision-makers. We note that Dasgupta et al. (2020) also uses the Kullback-Leibler function as a distance metric to be minimized. We comment on these papers in the Discussion of the revised manuscript (l. 804-827).

The authors cite a number of earlier modeling papers, but it's not clear to me what those previous models would predict about the new data, and whether the new models proposed in this paper predict the earlier data.

The main kind of other models that we refer to, in the Introduction and in the Discussion, are ‘leaky integration’ models, in which past observations are gradually forgotten (through an exponential discount). We show that the optimal solution to the problem of constrained inference (Equation (1)) with a precision cost (Equation (3)), is precisely one in which remote patterns in the sequence of observed stimuli are discounted, in the posterior, through an exponential filter. In other words, we recover the leaky-integration model (i.e., a model identical, for instance, to the one examined by Meyniel, Maheu and Dehaene, 2016), and thus the predictions of the precision-cost model are exactly those of a leaky-integration model (precision-cost models with different Markov order differ by the length of the sequences of observations that are counted through an exponential filter). One difference of our study, in comparison with previous works, is that the leaky integration is derived as the optimal solution to a problem of constrained optimization, rather than posited a priori in the definition of the model. We improved the revised manuscript, by clarifying in the Introduction that we recover leaky integration (l.76-79); by explaining in more details, in Results, that the precision-cost model results in a leaky-integration model, and by explicitly providing the posterior in the case of a Bernoulli observer (with the exponentially-filtered counts of past observations, Equation (6)); and by pointing out, in the Discussion, that we derive the exponential filtering from the constrained-inference problem, rather than assuming leaky integration from the start (l. 706-716).

Reference:

Florent Meyniel, Maxime Maheu, and Stanislas Dehaene. Human Inferences about Sequences: A Minimal Transition Probability Model. PLoS Computational Biology, 12(12):1–26, 2016.

Reviewer #2 (Recommendations for the authors):I would recommend the authors conduct additional modeling analyses including models that express the alternative hypotheses clearly, generate and present individual model predictions, and consider running a (possibly online) larger experiment with incentives in place. The ideal test of the hypothesis would be an experiment with multiple levels of incentives, showing that people adjust their representations as this model predicts as they trade off computational costs and real rewards. If monetary rewards cannot be used, incentives could be implemented by having a longer delay after erroneous predictions.

We thank Reviewer #2 for her/his attention to our paper and for her/his comments.

As for the point on the models: many models in the sequential-effects literature (Refs. [7-12] in the manuscript) are ‘leaky-integration’ models that interpret sequential effects as resulting from an attempt to learn the statistics of a sequence of stimuli, through exponentially decaying counts of the simple patterns in the sequence (e.g., single stimuli, repetitions, and alternations). In some studies, the ‘forgetting’ of remote observations that results from the exponential decay is justified by the fact that people live in environments that are usually changing: it is thus natural that they should expect that the statistics underlying the task’s stimuli undergo changes (although in most experiments, they do not), and if they expect changes, then they should discard old observations that are not anymore relevant. This theoretical justification raises the question as to why subjects do not seem to learn that the generative parameters in these tasks are in fact not changing — all the more as other studies suggest that subjects are able to learn the statistics of changes (and consistently they are able to adapt their inference) when the environment does undergo changes (Refs. [42,57]).

Our models are derived from a different approach: we derive behavior from the resolution of a problem of constrained optimization of the inference process. It is not a phenomenological model. When the constraint that weighs on the inference process is a cost on the precision of the posterior, as measured by its entropy, we find that the resulting posterior is one in which remote observations are ‘forgotten’, through an exponentially discount, i.e., we recover the predictions of the leaky-integration models, which past studies have empirically found to be reasonably good accounts of sequential effects. (Thus these models are already in our model comparison.) In our framework, the sequential effects do not stem from the subjects’ irrevocable belief that the statistics of the stimuli change from time to time, but rather from the difficulty that they have in representing precise belief; a rather different theoretical justification.

Furthermore, we show that a large fraction of subjects are not best-fitted by precision-cost models (i.e., they are not best-fitted by leaky integration), but instead they are best fitted by unpredictability-cost models. These models suggest a different explanation of sequential effects: that they result from the subjects favoring predictable environments, in their inference.

In the revised version of the manuscript, we have made clearer that the derivation of the optimal posterior under a precision cost results in the exponential forgetting of remote observations, as in the leaky-integration models. We mention it in the abstract, in the Introduction (l. 76-78), in the Results when presenting the precision-cost models (l. 264-278), and in the Discussion (l.706-716).

As for the point on incentivization: we agree that it would be very interesting to measure whether and to which extent the performance of subjects increases with the level of incentivization. Here, however, we wanted, first, to establish that subjects’ behavior could be understood as resulting from inference under a cost, and second, to examine the sensitivity of their predictions to the underlying generative probability — rather than to manipulating a tradeoff involving this cost (e.g. with financial reward). We note that we do find that subjects are sensitive to the generative probability, which implies that they exhibit some degree of motivation to put some effort in the task (which is the goal of incentivization), in spite of the lack of economic incentives. But it would indeed be interesting to know how the potential sensitivity to reward interacts with the sensitivity to the generative probability. Furthermore, as Reviewer #2 mentions, some studies show that incentives affect probability-matching behavior: it is then unclear whether the introduction of incentives in our task would change the inference of subjects (through a modification of the optimal trade-off that we model); or whether it would change their probability-matching behavior, as modeled by our generalized probability-matching response-selection strategy; or both. Note that we disentangled both aspects in our modeling and that our conclusions are about the inference, not the response-selection strategy. We deem the incentivization effects very much worth investigating; but they fall outside of the scope of our paper.

We now mention this point in the Discussion of the revised manuscript (l. 828-840).

Reviewer #3 (Recommendations for the authors):1. I recommend using regression analyses (e.g. a GLM) where regressors for both previous choices and previous stimuli are entered (as e.g. in Urai et al. Nature Communications 2017) to resolve this possible confound. Such GLM also allows looking back further back in time at the impact of longer lags on decisions. This would allow testing for example if the repulsive bias to the stimulus at lag 2 (in sequences 111 or 101) extends to longer lags, both in experimental data and simulations.

We thank Reviewer #3 for pointing out the possibility that subjects may have a tendency to repeat motor responses that is not related to their inference.

We note that in Urai et al., 2017, as in many other sensory 2AFC tasks, successive trials are independent: the stimulus at a given trial is a random event independent of the stimulus at the preceding trial; the response at a given trial should in principle be independent of the stimulus at the preceding trial; and the response at the preceding trial conveys no information about the response that should be given at the current trial (although subjects might exhibit a serial dependency in their responses). By contrast, in our task an event is more likely than not to be followed by the same event (because observing this event suggests that its probability is greater than.5); and a prediction at a given trial should be correlated with the stimuli at the preceding trials, and with the predictions at the preceding trials. In a logit model (or any other GLM), this would mean that the predictors exhibit multicollinearity, i.e., they are strongly correlated. Multicollinearity does not reduce the predictive power of a model, but it makes the identification of parameters extremely unreliable: in other words, we wouldn’t be able to confidently attribute to each predictor (e.g., the past observations and the past responses) a reliable weight in the subjects’ decisions. Furthermore, our study shows that past stimuli can yield both attractive and repulsive effects, depending on the exact sequence of past observations. To capture this in a (generalized) linear model, we would have to introduce interaction terms for each possible past sequence, resulting in a very high number of parameters to be identified.

However, this does not preclude the possibility that subjects may have a motor propensity to repeat responses. In order to take this hypothesis into account, we examined the behavior and the ability to capture subjects’ data of models in which the response-selection strategy allows for the possibility of repeating, or alternating, the preceding response. Specifically, we consider models that are identical to those in our study, except for the response-selection strategy, which is an extension of the generalized probability-matching strategy, in which a parameter eta, greater than -1 and lower than 1, determines the probability that the model subject repeats its preceding response, or conversely alternates and chooses the other response. With probability 1-|η|, the model subject follows the generalized probability-matching response-selection strategy (parameterized by κ). With probability |η|, the model subject repeats the preceding response, if η > 0, or chooses the other response, if η < 0. We included the possibility of an alternation bias (negative η), but we find that no subject is best-fitted by a negative η, thus we focus on the repetition bias (positive η). We fit the models by maximizing their likelihoods, and we compared, using the Bayesian Information Criterion (BIC), the quality of their fit to that of the original models that do not include a repetition propensity.

Taking into account the repetition bias of subjects leaves the assignment of subjects into two families of inference cost mostly unchanged. We find that for 26% of subjects the introduction of the repetition propensity does not improve the fit (as measured by the BIC) and can therefore be discarded. For 47% of subjects, the fit is better with the repetition propensity (lower BIC), and the best-fitting inference model (i.e., the type of cost, precision or unpredictability, and the Markov order) is the same with or without repetition propensity. Thus for 73% (=26+47) of subjects, allowing for a repetition propensity does not change the inference model. We also find that the best-fitting parameters λ and κ, for these subjects, are very stable, when allowing or not for the repetition propensity. For 11% of subjects, the fit is better with the repetition propensity, and the cost type of the inference model is the same (as without the repetition propensity), but the Markov order changes. For the remaining 16%, both the cost type and the Markov order change.

Thus for a majority of subjects, the BIC is improved when a repetition propensity is included, suggesting that there is indeed a tendency to repeat responses, independent of the subjects’ inference process and generative stimulus probability. In Figure 7, in Methods, we show the behavior of the models without repetition propensity, and with repetition propensity, with a parameter η = 0.2 close to the average best-fitting value of eta across subjects. We show, in Methods, that (i) the unconditional probability of a prediction A, p(A), is the same with and without repetition propensity, and that (ii) the conditional probabilities p(A|A) and p(A|B) when η≠0 are weighted means of the unconditional probability p(A) and of the conditional probabilities when eta=0 (see p. 47-49 of the revised manuscript).

In summary, our results suggest that a majority of subjects do exhibit a propensity to repeat their responses. Most subjects, however, are best-fitted by the same inference model, with or without repetition propensity, and the parameters λ and κ are stable, across these two cases; this speaks to the robustness of our model fitting. We conclude that the models of inference under a cost capture essential aspects of the behavioral data, which does not exclude, and is not confounded by, the existence of a tendency, in subjects, to repeat motor responses.

In the revised manuscript, we present this analysis in Methods (p.47-49), and we refer to it in the main text (l. 353-356 and 400-406).

2. Authors could test whether the very first prediction on each block already shows a signature of the p and whether the prediction is stable within blocks.

The assumptions that subjects reach their asymptotic behavior after being presented with 200 observations in the passive trials should indeed be tested. To that end, we compared the behavior of the subjects in the first 100 active trials with their behavior in the remaining 100 active trials. The results of this analysis are shown in figure 9.

For most values of the stimulus generative probability, the unconditional proportions of predictions A, in the first and the second half (panel a, solid and dashed gray lines), are not significantly different (panel a, white dots), except for two values (p-value < 0.05; panel a, filled dots). Although in most cases the difference between the two is not significant, in the second half the proportions of prediction A seem slightly closer to the extremes (0 and 1), i.e., closer to the optimal proportions. As for the sequential effects, they appear very similar in the two halves of trials. We conclude that for the purpose of our analysis we can reasonably consider that the behavior of the subjects is stationary throughout the task.

On top of that, I provide here some recommendations on how the clarity of the manuscript could be improved. A thorough work on improving the text and figures and the general flow and organization of the manuscript would make a major difference in the impact of that paper on the wider community. which is very frustrating for the reader is to see one statement (e.g. a choice of methods or a result) exposed at some point and then the explanation for the statement much later in the manuscript (see below). Here are my suggestions:– I believe the models would be better motivated to the reader if the Introduction made a brief mention of the ideas of bounded rationality (and related concepts) and justified focus on these two specific types of cost – all of which are nicely detailed in the Discussion.– Please try to make understanding figures more intuitive; for example, using a colour code for the different cost types may help differentiate them. A tree-like representation of history biases (showing the mean accuracy for different types of sequences, e.g. in Meyniel et al. 2016 Plos CB) may be more intuitive to read and reveal a richer structure in the data and models than the current Figure 5-6 (also given than the authors do not comment much on the impact of the "probability of observation 1", so perhaps this effects could be marginalized out).– Figure 3 is really helpful in understanding the two types of cost (much more than the equations for most readers). Unfortunately, it is hardly referred to in the main text. I suggest rewriting the presentation of that part of the Results section around these examples.– Why and how the two types of costs give rise to historical effects (beyond the fact that these costs generate suboptimalities) is a central idea in the paper but it is only exposed in the Discussion session. Integrating these explanations within the Results section would help a lot. Plotting some example simulations for a sequence of trials and/or some cartoon explanations of how the historical effects emerge for the different models would also help.– Placing figures in Methods does not help in my opinion – please consider moving to the main text or as supplementary Figures.

We thank Reviewer #3 for these recommendations, which we have followed in the revised manuscript.

– We now explain early in the Introduction how our approach relates to other “resource-rational” accounts of perceptual and cognitive processes, in which behavior is assumed to result from some constrained optimization. We also provide more details on the two costs we consider in the paper (l. 65-79).

– As for the color code for the different costs, the colors used for each cost in Figure 3 are consistent across panels. In the other figures, we have favored a color-coding that emphasizes the sequential effects (allowing to distinguish, for instance, p(A|A) and p(A|B)). However, in the revised manuscript, wherever a figure shows a simulation resulting from one of the two costs, we have set the color of the figure’s axes, and of the diagonal (\bar p = p), to the color that corresponds to the cost (consistently with Figure 3), so as to facilitate the identification of the models across figures.

– We have now included a “tree-like” representation of the sequential effects, in Figure 6d of the revised manuscript. The impact of the stimulus generative probability was “marginalized out”, as suggested by Reviewer #3. The nonmarginalized tree representations (per stimulus probability) can be found in Methods (Figure 13).

– We now refer more often to Figure 3, in the Results section of the main text (p. 11-15), so as to connect what is described in the text (e.g., the nonconvergence of the posterior with the precision cost) to its illustration in Figure 3. In addition we also refer to it in the section describing the sequential effects of the model, so as to make clear that with the precision-cost models the sequential effects stem from the non-convergence of the posterior (p. 20).

– In the revised manuscript we give more detailed explanations as to how each cost gives rise to sequential effects (which we agree is an important conceptual idea in the paper). In particular we have revised the passages in the Results section in which we present these costs and the corresponding solutions to the optimization problem (p. 11-15). Furthermore, we have added a new panel in Figure 3 (panel d), which shows, as suggested by the reviewer, the “trajectories” of a subject’s estimates, for different sequences of observations (though with the same stimulus generative probability), under the two costs, and under no cost (the Bayesian solution). It shows that the Bayesian observer converges to the correct value of the stimulus generative probability; that the observer under an unpredictability cost converges to an erroneous value; and that the observer under a precision cost does not converge, but keeps fluctuating with the history of stimuli. Finally, when describing the sequential effects of the model, we explain in detail how and why each cost gives rise to the sequential effects shown in Figure 4.

– We have reconsidered the Figures in Methods and whether we should place them elsewhere. We now place in Supplementary Materials the modelfitting confusion matrix and the figure showing the stability of the costweight parameter across medium and extreme values of the stimulus generative probability. We have kept the two figures (Figures 7 and 8) showing the sequential effects of each of the eight models (with each cost type and each Markov order) in Methods, because we think that they provide interesting information and that they are probably expected by the reader, although it is not crucial that they appear in the flow of the main text.